# A tipping point in refreezing accelerates mass loss of Greenland's glaciers and ice caps

B. Noël[1], W.J. van de Berg[1], S. Lhermitte[2], B. Wouters[1], H. Machguth[3,4,5], I. Howat[6], M. Citterio[5], G. Moholdt[7], J.T.M. Lenaerts[1] & M.R. van den Broeke[1]

Melting of the Greenland ice sheet (GrIS) and its peripheral glaciers and ice caps (GICs) contributes about 43% to contemporary sea level rise. While patterns of GrIS mass loss are well studied, the spatial and temporal evolution of GICs mass loss and the acting processes have remained unclear. Here we use a novel, 1 km surface mass balance product, evaluated against *in situ* and remote sensing data, to identify 1997 ($\pm$ 5 years) as a tipping point for GICs mass balance. That year marks the onset of a rapid deterioration in the capacity of the GICs firn to refreeze meltwater. Consequently, GICs runoff increases 65% faster than meltwater production, tripling the post-1997 mass loss to $36 \pm 16$ Gt$^{-1}$, or $\sim$14% of the Greenland total. In sharp contrast, the extensive inland firn of the GrIS retains most of its refreezing capacity for now, buffering 22% of the increased meltwater production. This underlines the very different response of the GICs and GrIS to atmospheric warming.

[1] Institute for Marine and Atmospheric Research Utrecht, Utrecht University, 3584 CC Utrecht, The Netherlands. [2] Department of Geoscience and Remote Sensing, Delft University of Technology, 2600 AA Delft, The Netherlands. [3] Department of Geography, University of Zürich, CH-8006 Zürich, Switzerland. [4] Department of Geosciences, University of Fribourg, CH-1700 Fribourg, Switzerland. [5] Geological Survey of Denmark and Greenland GEUS, 1350 København K, Denmark. [6] Byrd Polar Research Center and School of Earth Sciences and Department of Geography, Ohio State University, Columbus, Ohio 43210, USA. [7] Norwegian Polar Institute, Fram Centre, NO-9296 Tromsø, Norway. Correspondence and requests for materials should be addressed to B.N. (email: b.p.y.noel@uu.nl).

Covering a total area of ~90,000 km², Greenland's peripheral glaciers and ice caps (GICs) represent ~12% of the world's glacierized area outside of the Antarctic and Greenland ice sheets[1]. Greenland's GICs account for 14 to 20% of total current Greenland glacial mass loss[2], although they only represent ~5% of the area and ~0.5% (~39 mm SLE) of the volume of the Greenland ice sheet (GrIS). In a scenario of continued global warming, Greenland's GICs may lose 19–28% (7.5–11 mm) of their volume by 2100 (ref. 3). Despite multiple *in situ* observational campaigns since the early 1950s (ref. 4), glacier modelling[5] and satellite-based[2,6] estimates, large uncertainties remain in the spatial and temporal distribution of Greenland's GICs mass loss. To fill these gaps, regional climate models (RCMs) are often used[7–15], but their horizontal resolution (typically 5–20 km) fails to resolve the steep surface mass balance (SMB) gradients in the topographically complex regions in which GICs are often situated[16]. To address this issue, we created a 1 km data set, statistically downscaled from output of the regional atmospheric climate model RACMO2.3 using regressions of SMB components against elevation estimated at the model resolution of 11 km. These regressions are then applied to a downsampled 1 km version of the topography and ice mask of the Greenland Ice Mapping Project (GIMP) Digital Elevation Model (DEM)[17]. The downscaling procedure also includes a bare ice albedo correction based on a 1 km MODIS albedo product to avoid underestimation of melt and runoff, especially on dark, low-lying glacier tongues. Earlier, the downscaling method was successfully applied to the GrIS[16].

Here we use the novel SMB product at 1 km resolution to quantify Greenland's GICs mass loss, assuming changes in solid ice discharge to be negligible[18–20]. The data set includes individual SMB components (precipitation, sublimation, melt, refreezing and runoff) for all GICs on a daily time scale (1958–2015), which is crucial for evaluation using irregular (in time and space) observations and to understand the drivers of mass loss. Using this product, we identify 1997 (± 5 years) as a tipping point for the mass balance of Greenland's GICs, which marks the onset of a rapid deterioration of inland firn capacity to refreeze meltwater, causing long-term mass loss.

## Results

**Evaluation against observations.** Figure 1a shows average (1958–2015) downscaled SMB for the whole of Greenland, and Fig. 1b compares the downscaled SMB to 965 SMB measurements from 101 GICs sites[4] (yellow dots in Fig. 1a). With 77% of the variance explained, the downscaled SMB agrees well with observations, although significant deviations and a negative bias of 240 mm w.e. yr⁻¹ (water equivalent) remain. We selected five GICs regions (black boxes) to highlight the agreement and differences of the downscaled SMB product with observations (Fig. 1a). Supplementary Figure 1 shows and briefly discusses the intricate patterns of SMB and its components over GICs in four of these regions (Supplementary Discussion); sector five is discussed below.

A direct comparison with mass loss estimates from independent ICESat/CryoSat-2 measurements over the period 2004–2015 demonstrates that the downscaled SMB product with albedo correction successfully reproduces GICs mass changes (Fig. 2), including seasonal and interannual variability, for example, the large difference in mass loss between the summers of 2012 and 2013. The uncertainty in downscaled SMB was estimated at 15.7 Gt yr⁻¹ (~40%, see Methods). Note that the Greenland's GICs area of ~81,400 km² used in this study is smaller by ~8% than previous estimates[21,22] due to the omission of unresolved small ice bodies (<1 km²) in the original GIMP DEM.

**A tipping point in GICs mass balance.** Figure 3 shows time series (1958–2015) of annual mean SMB components precipitation (PR), melt (ME), refreezing (RF) and runoff (RU), all in Gt yr⁻¹ and spatially integrated over (a) Greenland's GICs and (b) the GrIS. Figure 3c zooms in on the refreezing time series. Using a breakpoint analysis (see Methods), we identify 1997 (± 5 years) as the year after which the GICs refreezing regime starts to decrease and diverges significantly from the GrIS refreezing regime (black point in Fig. 3c). This marked reduction in refreezing capacity is representative of a deteriorating firn layer, the porous, multiyear snow layer between surface fresh snow (~350 kg m⁻³) and the underlying ice (~900 kg m⁻³). Decades of increased melt have reduced pore space to such a degree that enhanced refreezing can no longer compensate for increased meltwater production. Because it would take decades to regrow a healthy firn layer, we interpret 1997 as a tipping point in the mass balance of Greenland's GICs.

## Discussion

Prior to 1997, Greenland's GICs average SMB was marginally negative (− 11.3 ± 15.7 Gt yr⁻¹; see Supplementary Tables 1 and 2 for numbers in mm w.e. yr⁻¹), with an insignificant trend of − 0.01 ± 0.22 Gt yr⁻². Note how, even in this earlier period, melt (95 Gt yr⁻¹) persistently and significantly exceeded precipitation (58 Gt yr⁻¹), stressing the importance of the refreezing process for maintaining the mass balance of these Arctic GICs close to zero (Supplementary Table 1). The situation is very different for the GrIS, where pre-1997 precipitation (746 Gt yr⁻¹) exceeded melt (556 Gt yr⁻¹) by a wide margin (Supplementary Table 2). Coincidentally, before 1997 the GICs and the GrIS had a similar refreezing fraction (RF/ME) of 38% and 43%, respectively.

Between 1997 and 2015, the integrated GICs SMB decreased at a rate of 1.1 ± 0.6 Gt yr⁻², signifying mass loss acceleration, resulting in an average SMB over this period of − 36.2 ± 15.7 Gt yr⁻¹. Previous estimates for different periods[2,5,6] (Supplementary Table 3) confirm this recent increase of GICs mass loss. However, with the new SMB product we are now able to identify the physical processes responsible for the post-1997 mass loss acceleration. Figure 3a unambiguously shows that the trend in SMB is almost exclusively driven by increased runoff (1.1 ± 0.6 Gt yr⁻²), while precipitation remains constant. A similar pattern emerges for the GrIS; here, a negative trend in precipitation (− 3.5 ± 2.6 Gt yr⁻²) somewhat reinforces the decrease in SMB (− 10.4 ± 4.0 Gt yr⁻²), but again the latter is dominated by the increase in runoff (6.9 ± 3.7 Gt yr⁻²). But Fig. 3 and Supplementary Tables 1 and 2 also reveal a striking difference in the responses of the GrIS and Greenland's GICs to atmospheric warming. On the GrIS (Fig. 3b and Supplementary Table 2), an important fraction (22%) of the excess meltwater produced since 1997 has been retained in the extensive interior firn layer, driven by an increase in refreezing (RF). In contrast, refreezing decreased on Greenland's GICs (Fig. 3c). As a result, runoff outpaces excess meltwater production by 65% since 1997. It thus appears that the mass balance of Greenland's GICs crossed a tipping point in 1997 (Fig. 3c), implying eventual long-term loss of the firn layer's refreezing capacity.

Figure 4 confirms these findings by comparing vertical profiles of surface mass fluxes integrated over GICs and GrIS elevation bins, scaled by the maximum height per region ($h_{max}$), prior to and after 1997. Figure 4a,b shows that the equilibrium line (SMB = 0) of the GICs moved significantly upward, that is, from 0.61 to 0.71 of $h_{max}$, and is now situated well above the peak in the hypsometry (0.62, Fig. 4d). In combination with decades of increased melt, which depleted firn pore space, the GICs firn layer

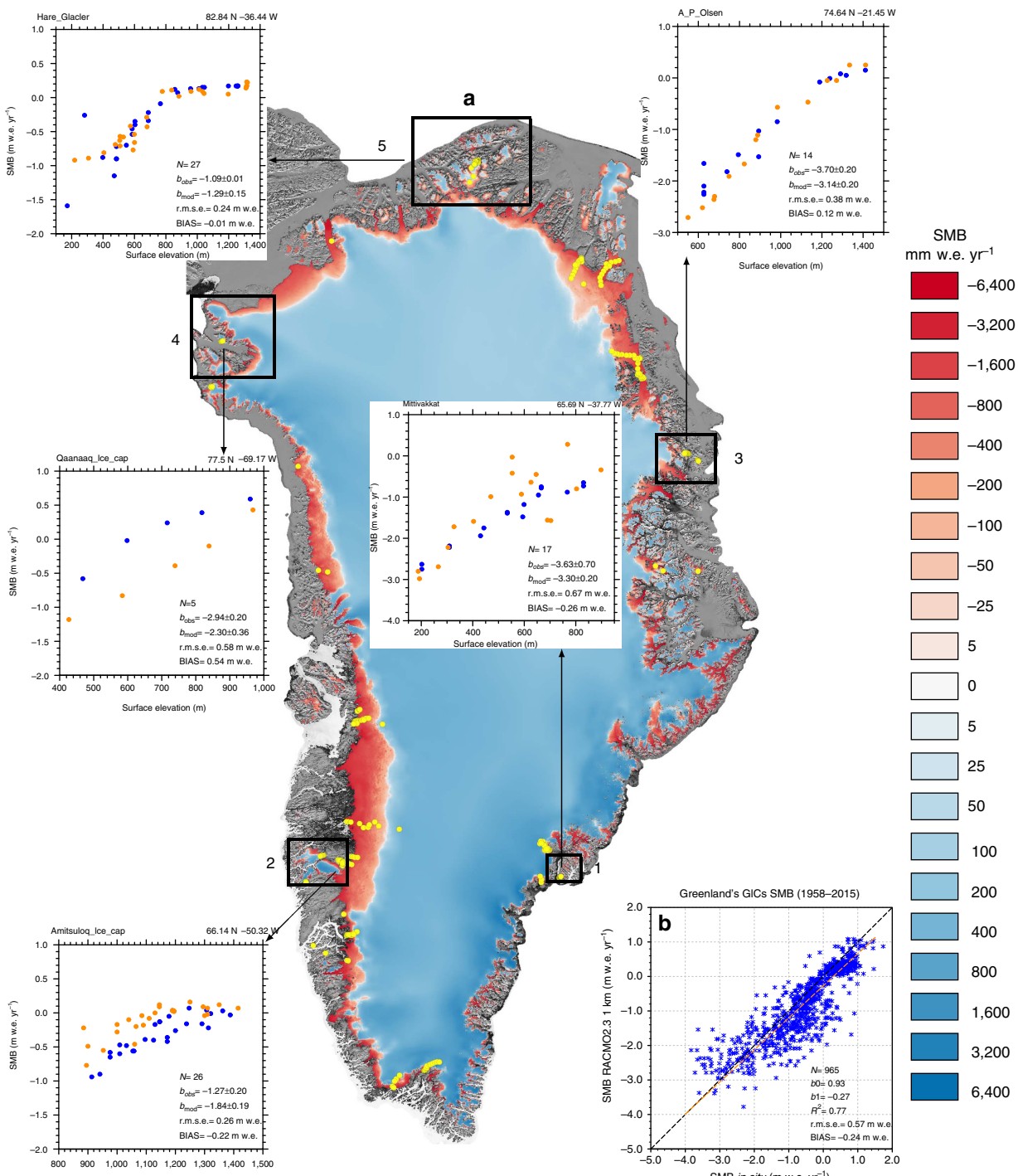

**Figure 1 | Greenland SMB patterns and evaluation.** (**a**) Annual mean downscaled SMB (v1.0, see Methods) at 1 km resolution over the GrIS and neighbouring GICs for 1958–2015. Yellow dots correspond to 331 SMB observation sites used for GrIS (230) and GICs (101) evaluation. Numbered black boxes depict five regions including stake transects collected over GICs. For each transect, annual mean SMB is plotted from downscaled simulation v1.0 ($b_{mod}$; dark blue dots) and *in situ* data ($b_{obs}$; orange dots). The number of observations, observed and downscaled SMB gradients, r.m.s.e. and mean BIAS are also listed for each transect. (**b**) Comparison of ablation measurements collected at 101 GICs sites with downscaled SMB at 1 km (v1.0). The orange dashed line represents the regression including all measurements ($y = b_1 + b_0 \times x$).

is no longer capable of buffering the excess meltwater production. As a result, runoff increases at the same rate as melt and fully governs the GICs mass loss (Fig. 4c). Figure 4a,b also shows that rainfall (RA) is a small (6%) fraction of the liquid water flux available at the firn layer top, which is dominated by melt. For the GrIS the equilibrium line has moved upwards from 0.33 to 0.40 of

$h_{max}$ (Fig. 4e,f), but remains well below the maximum in the GrIS hypsometry (0.73, Fig. 4h). Therefore, a significant part of the excess melt is buffered by refreezing (Fig. 4g) and runoff remains constant above 0.61 of $h_{max}$. Although formation of ice lenses may reduce the retention efficiency in the lower accumulation zone[23,24], we conclude that the extensive and elevated inland firn

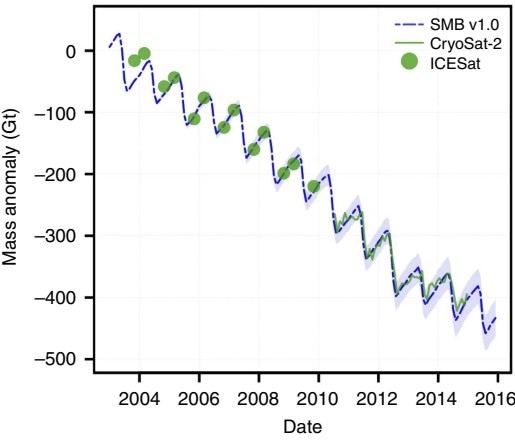

**Figure 2 | Contemporary Greenland's GICs mass anomaly.** Comparison of Greenland's GICs-integrated mass anomaly (2004–2015) derived from ICESat (green dots)/CryoSat-2 (green line) altimetry with the downscaled SMB v1.0 (dashed blue line). The light blue belt represents the estimated uncertainty of the downscaled SMB product.

area of the GrIS (Fig. 4h) maintains its refreezing capacity for now. As a result, the acceleration of GrIS surface mass loss is less than half that of the GICs ($6.1 \pm 2.4$ mm w.e. yr$^{-2}$ versus $13.5 \pm 7.4$ mm w.e. yr$^{-2}$).

To analyse spatial differences, we examined the changes in runoff, refreezing fraction (RF/ME) and 2-m air temperature in 12 marginal regions of Greenland (1997–2015 minus 1958–1996), covering 85% of all Greenland's GICs (Supplementary Fig. 2). All regions experienced warming and an increase in runoff, but a marked contrast is found between GICs in north and south Greenland. Northern GICs experienced a significantly greater warming than southern GICs ($+1.0$ to $1.5\,°C$, orange boxes versus $+0.6$ to $0.8\,°C$, black boxes)[25,26] and a larger relative increase in runoff ($+50$ to 74% versus $+17$ to 34%). The reduction in refreezing fraction is also twice as large for northern than for southern GICs (9 to 14% versus 5 to 7%; Supplementary Fig. 2). To further investigate the mechanisms involved, Fig. 5a zooms in on the region of Hans Tausen ice cap in north Greenland (region 5 in Fig. 1a). The Hans Tausen region shows a small steady mass loss ($2.6 \pm 1.4$ Gt yr$^{-1}$) before 1997 and a tripling in mass loss ($7.6 \pm 1.4$ Gt yr$^{-1}$) afterwards (Supplementary Fig. 3). Figure 5b shows that the change in runoff (1997–2015 minus 1958–1996) is largest in the narrow ablation zone along the margins (300–500 mm w.e. yr$^{-1}$), but with a significant contribution from the interior ($\sim 100$ mm w.e. yr$^{-1}$). This implies that all but the very highest parts of the GICs accumulation zones now regularly experience runoff, that is, the firn layer is no longer capable of refreezing all meltwater that is produced in summer. Figure 5c confirms that the upper firn area of northern Greenland GICs has experienced the largest negative changes, up to 50%, in the refrozen meltwater fraction (RF/ME). Figure 5d relates these changes, as well as changes in rainfall, to annual mean (downscaled) 2-m air temperature anomalies. The strong correlation proves that the recent warming has reduced the refreezing capacity of the firn layer in these high northern GICs: in warm years, enhanced dry snow densification and surface melt quickly saturate the pore space of the firn layer. In addition, the fraction of rainfall doubles from $\sim 5$ to $\sim 10\%$ in warm compared to cold years, further limiting the formation of firn. The resulting reduced refreezing capacity means that continuous warming in the future is likely to further accelerate Greenland's GICs mass loss and rapidly erode these highly sensitive northern ice masses.

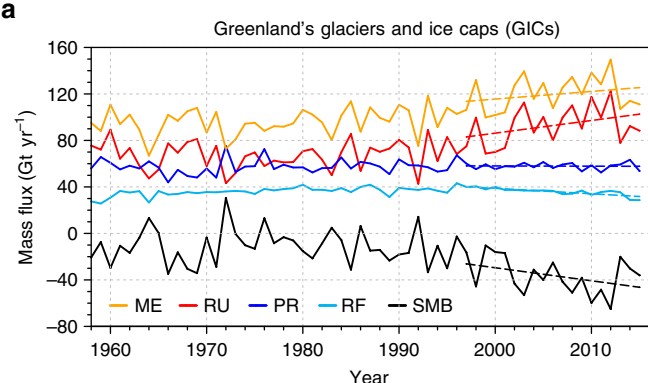

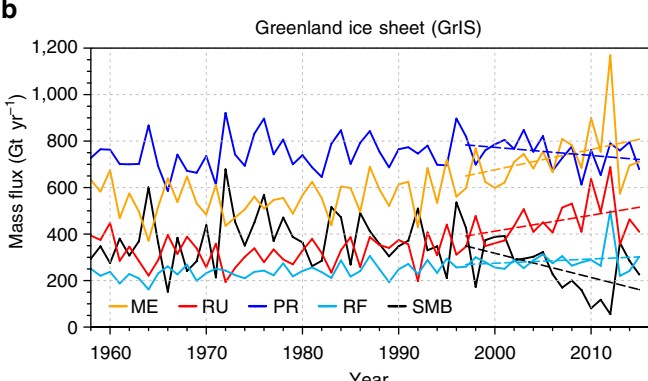

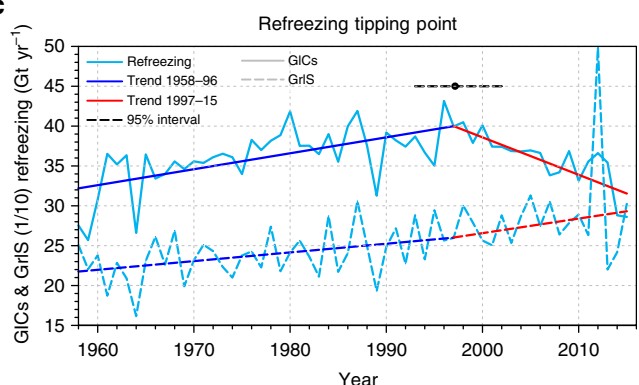

**Figure 3 | Mass flux evolution and refreezing tipping point.** (**a**) Time series of GICs annual cumulative SMB components for the period 1958-2015. (**b**) Same as **a** but for the contiguous GrIS. Dashed lines show 1997–2015 trends. Total sublimation (SU) and snow drift erosion (ER) are not included in the above time series as they contribute relatively little to SMB and trends are very small compared to the other components. (**c**) Time series of GICs (continuous lines) and GrIS (1/10, dashed lines) integrated annual mean refreezing and trends for 1958–1996 (blue) and 1997–2015 (red). The GICs refreezing tipping point (1997) is represented in black with a 95% confidence interval (dashed black line).

## Methods
**Regional climate model.** Output of the Regional Atmospheric Climate Model (RACMO2.3) is used as input for the downscaling procedure[14,16]. RACMO2.3 combines the atmospheric dynamics from the High-Resolution Limited Area Model (HIRLAM) and the physics from the European Centre for Medium-range Weather Forecasts Integrated Forecast System (ECMWF-IFS)[27]. The polar version of RACMO2.3 is developed by the Institute for Marine and Atmospheric Research Utrecht University (IMAU), to simulate the evolution of SMB over ice sheets and surrounding smaller glacierized regions. Polar RACMO2.3 incorporates a multi-layer snow module to simulate firn compaction, meltwater retention and percolation, refreezing and runoff[9]. In RACMO2.3, the excess energy available at

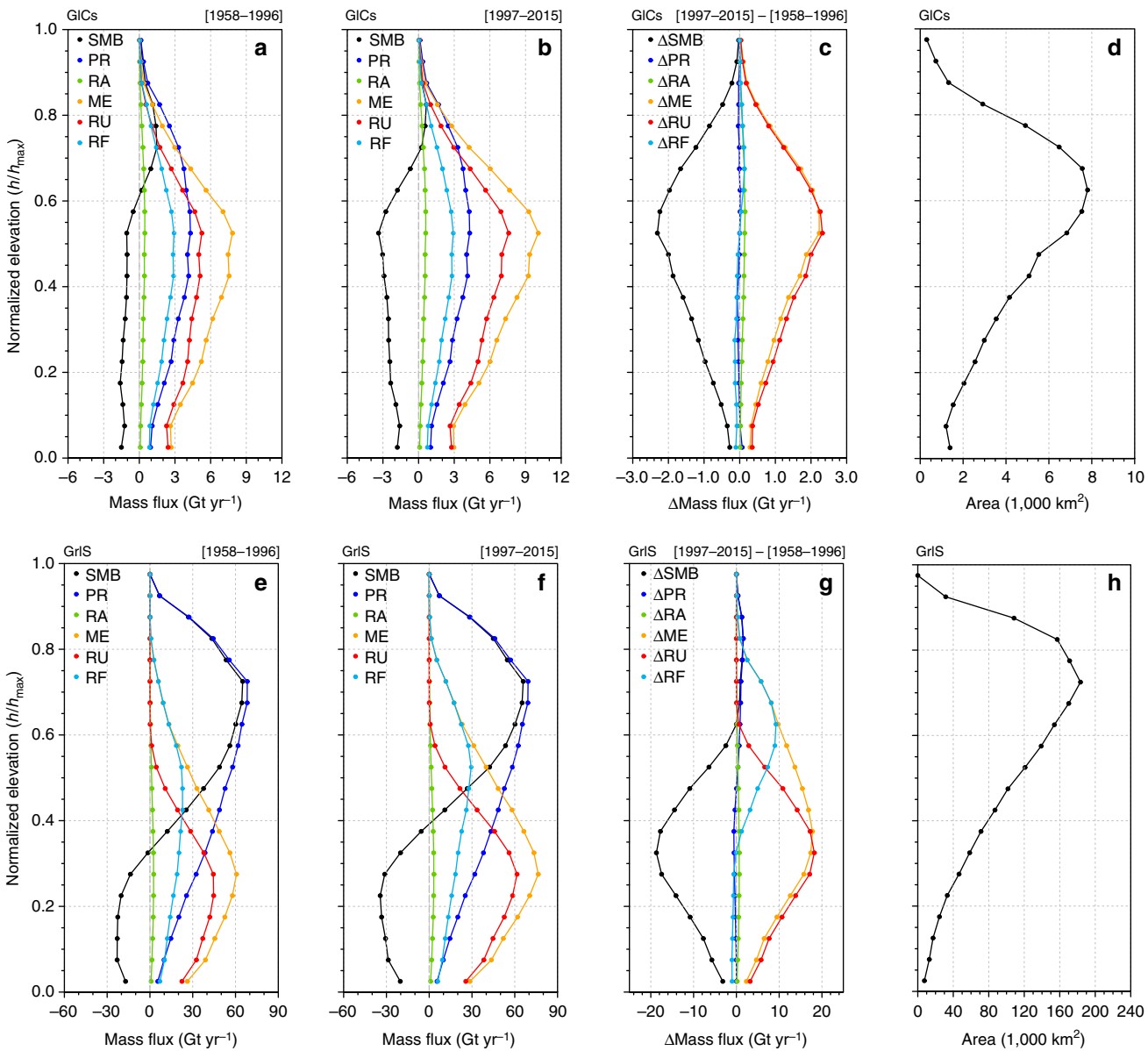

**Figure 4 | Shrinking of the accumulation zones.** Vertical profiles of surface mass fluxes integrated over GICs (upper row) and GrIS (lower row) elevation bins, scaled by the maximum height per region ($h_{max}$), for the period 1958–1996 (**a,e**), 1997–2015 (**b,f**) and the difference between the two periods (1997–2015 minus 1958–1996, **c,g**). SMB components are spatially integrated within normalized elevation bins ($h/h_{max}$) of magnitude 0.05. For the GICs, SMB components are first integrated in elevation bins over twelve individual regions, each with different $h_{max}$ (boxes in Supplementary Fig. 5); the GICs-integrated SMB components are obtained by summing the contribution of the 12 regions to each scaled elevation bin. (**d,h**) The scaled hypsometries, that is, total area occupied by each elevation bin, for the GICs and the GrIS, respectively.

the surface, resulting from closure of the surface energy budget, is used to melt snow and ice. Liquid water from melt and rain percolates through the firn column, and is either held as irreducible water or refreezes, progressively reducing pore space from bottom to top layers until the entire firn column turns into ice (900 kg m$^{-3}$) and no additional water can be stored. At this point, any additional water is assumed to run off. The model also includes a snow albedo scheme using prognostic snow grain size[28]; a drifting snow routine accounting for sublimation and snow erosion[29]. For the contemporary Arctic simulation, RACMO2.3 was run at 11 km and forced on a six-hourly basis by ERA-40 (ref. 30; 1958–1978) and ERA-Interim[31] (1979–2015) re-analyses. The ice mask and topography at 11 km are based on a 5 km Greenland DEM[32]. For more information about RACMO2.3, recent updates and evaluation see refs 14,33,34.

**Topography and ice mask.** We used a downsampled version of the GIMP DEM[17] to downscale the output of RACMO2.3 to a 1 km topography and ice mask (Supplementary Fig. 4a,b), obtained by averaging the original GIMP DEM at 90-m resolution. To distinguish between the ice sheet, including glaciers strongly connected to the ice sheet (corresponding to connectivity level CL2 in ref. 21), and

GICs that are physically or dynamically separated from the ice sheet (respectively CL0 and CL1 in ref. 21), we used the Programme for Monitoring of the Greenland Ice Sheet (PROMICE) ice classes projected on the 1 km GIMP ice mask (Supplementary Fig. 4b). In addition, floating glacier tongues were eliminated using a 1 km ice grounding line[35]. This results in a GICs area of ∼81,400 km$^2$, ∼8% less than previous estimates[22], owing to unresolved small glaciers in the original GIMP DEM at 90-m (ref. 17).

**Bare ice albedo.** To correct for the systematic bare ice albedo overestimation of RACMO2.3 in low-lying glaciated regions at 11 km, we used a 1 km version of the 500 m MODerate-resolution Imaging Spectroradiometer (MODIS) 16-day Albedo product (MCD43A3). Bare ice albedo is defined as the average of the 5% lowest surface albedos recorded for 2000–2015. At 1 km, bare ice albedo values range from 0.15 for dark ice surfaces at the ablation zone edges and local glacier tongues, to 0.55 under persistent snow cover in the accumulation zone[16] (Supplementary Fig. 4c). In RACMO2.3, bare ice albedo is prescribed from a similar 11 km product (2001–2010) with a lower threshold of 0.30 (ref. 14). A value of 0.47 is assigned to glaciated pixels showing no valid MODIS estimate.

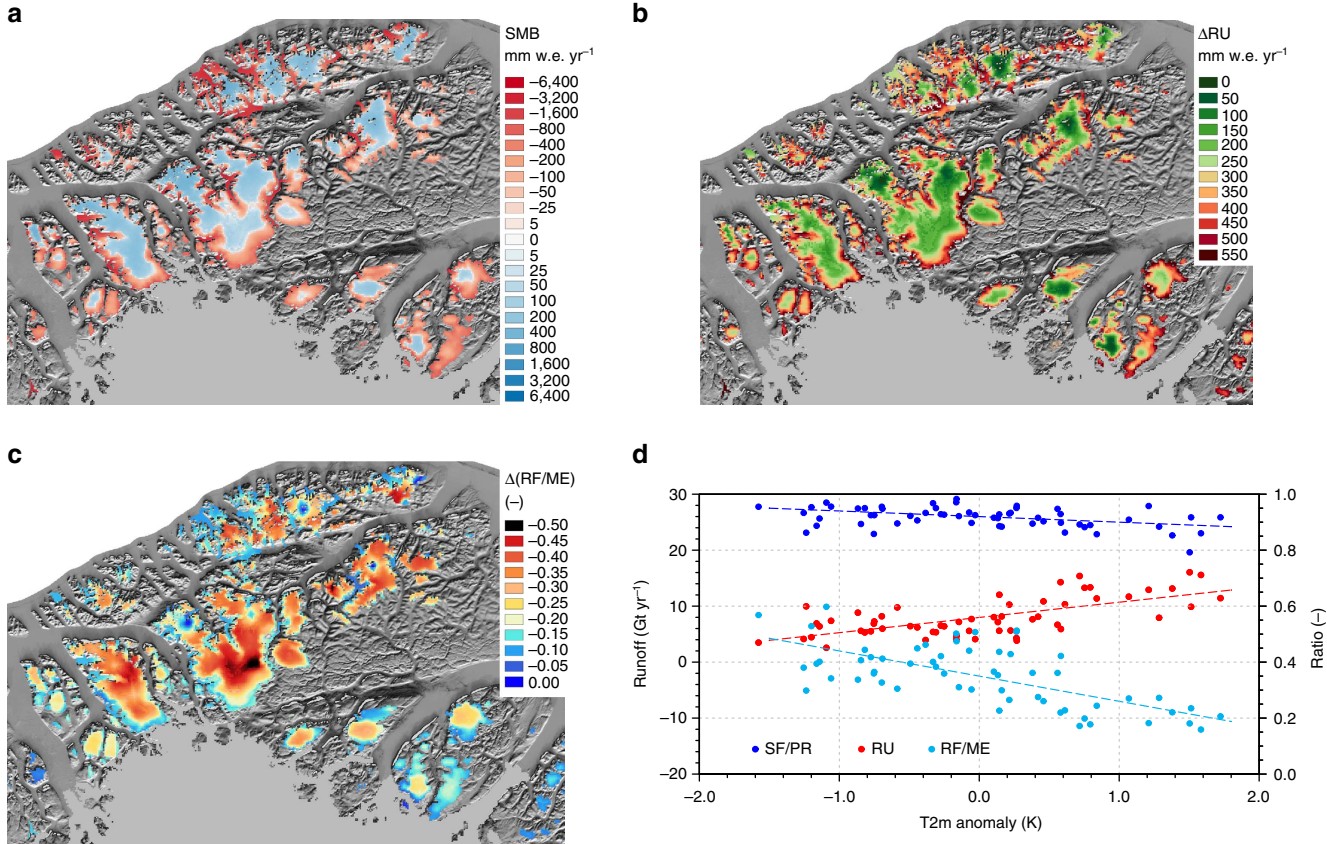

**Figure 5 | Mass loss amplification in the Hans Tausen region.** (**a**) Annual mean SMB patterns over the Hans Tausen region for 1958–2015 (black box 5 in Fig. 1), (**b**) difference in annual runoff between 1997–2015 and 1958–1996, (**c**) difference in the annual mean refreezing-to-melt ratio between 1997–2015 and 1958–1996 and (**d**) annual runoff (red), snowfall-to-precipitation ratio (dark blue) and refreezing-to-melt ratio (light blue) as a function of the anomaly in annual mean 2-m temperature.

**In-situ measurements.** A total of 965 SMB measurements collected at 101 stake sites[4] was used to evaluate the downscaled GICs SMB product at 1 km. These records were combined with another 1,073 observations from 230 sites on the GrIS (yellow dots in Fig. 1a) to adjust runoff and melt in the downscaling procedure. The ablation data set[4] was compiled as part of the PROMICE[36]. For consistency, we only selected data with a temporal overlap with RACMO2.3 (1958–2015), and rejected sites with a >100 m height difference relative to the GIMP DEM at 1 km. To compare downscaled and observed SMB (Fig. 1b), we selected the grid cell with the smallest elevation bias among the closest pixel and its eight neighbours.

**Remote sensing.** Elevation changes for 2003–2009 and 2010–2014 were derived from ICESat and CryoSat-2 measurements, following the methods described in refs 6,37. For ICESat, observations were grouped every 700 m along repeated ground tracks, whereas for CryoSat-2, neighbouring observations are collected within 1 km of each individual echo location. To these clusters of elevation observations, a model is fitted to estimate the local surface topography and elevation rate at the central point, where outliers are removed in an iterative procedure. For full details, see ref 37. After estimating the local topography and elevation rate for the ICESat and CryoSat-2 periods, local elevation anomalies at the echo locations can be estimated by adding back the constant elevation rate of the fitted model to the residuals. These anomalies are subsequently used to compute monthly volume anomalies for selected regions. We do so by parameterizing the elevation anomalies as a function of absolute elevation using a third-order polynomial. The resulting fit is then used to derive regional volume anomalies within 100 m elevation intervals, by multiplying the polynomial value at each interval's midpoint with the total glaciated area within this elevation bin[38]. We perform the polynomial fit for nine regions individually[6] and sum the results to obtain the GICs volume anomaly. Finally, volume anomalies are converted to mass anomalies by assuming a constant density profile, using the density of ice below the equilibrium line altitude (ELA), and a density of 600 ± 250 kg m$^{-3}$ above the ELA. Figure 2 shows the cumulative GICs mass change; using cumulative values suppresses noise in the ICESat and CryoSat-2 time series.

**Downscaling procedure.** The daily, 1 km SMB product is statistically downscaled from the output of RACMO2.3 at 11 km resolution (1958–2015) to the 1 km GIMP topography and ice mask (Supplementary Fig. 4a,b), using elevation dependence. Elevation correction is only applied to SMB components showing a significant correlation with height: runoff (RU), melt (ME) and sublimation (SU)[16]; total precipitation (PR), that is, rainfall (RA) and snowfall (SF), and snowdrift erosion (ER) are bi-linearly interpolated to the 1 km ice mask, without elevation correction. Daily SMB is then reconstructed as:

$$SMB = PR - RU - SU - ER \tag{1}$$

**Elevation dependence.** Daily regression parameters are calculated for the dependence of modelled SMB components on the 11 km RACMO2.3 topography. A local regression slope, $b_{11\,km}$ (mm w.e. per m, Supplementary Fig. 5), is estimated for each glacierized 11 km pixel using at least five adjacent ice-covered pixels. By applying the obtained $b_{11\,km}$ to the current pixel, the SMB component is approximated at mean sea level, $a_{11\,km}$ (mm w.e., Supplementary Fig. 5). To fully cover the GrIS and detached GICs, the latter regression parameters are extrapolated outwards by averaging $b_{11\,km}$ from at least three glaciated pixels. An estimate of $a_{1\,km}$ and $b_{1\,km}$ is then obtained by interpolating bi-linearly the 11 km regression parameters to the 1 km ice mask. Runoff, melt and sublimation ($X_{v0.2}$) are calculated using a linear function of the high-resolution topography ($h_{1\,km}$) as:

$$X_{v0.2} = a_{1\,km} + b_{1\,km} \times h_{1\,km} \tag{2}$$

The downscaled product based on elevation dependence only is hereafter referred to as version v0.2.

**Runoff and melt adjustments.** To correct for bare ice albedo overestimation in RACMO2.3 at 11 km, melt and runoff (v0.2) are adjusted to account for additional ice melt (ME$_{add}$) observed in low-lying regions compared to the downscaled product v0.2.

$$ME_{add} = \Delta\alpha \times 0.5 \times \frac{SW_{direct\,1\,km}}{L_f} \times (1 + \xi) \tag{3}$$

where ME$_{add}$ (mm w.e. per day) is the additional amount of ice melt calculated at 1 km (Supplementary Fig. 4c); $\Delta\alpha$ (dimensionless) is the difference between the averaged bare ice albedo retrieved from the set of regression cells used to downscale runoff at 11 km and the MODIS albedo product at 1 km; SW$_{direct\,1\,km}$ is the

modelled daily cumulated downward shortwave radiation bi-linearly interpolated to 1 km; $L_f$ is the latent heat of fusion ($3.337 \times 10^5 \, \mathrm{J\,kg^{-1}}$). To account for the slope of the glacier surface, the dimensionless correction factor for a tilted plane $\xi$ is applied to the direct component of modelled downward shortwave radiation. This correction is required as RACMO2.3 models radiation assuming a horizontal surface. This factor ranges from 0 for north sloping glaciers to values larger than 1 for south-oriented slopes.

A daily ratio $\Gamma$, between downscaled runoff and melt in v0.2, is applied to $\mathrm{ME_{add}}$ to calculate the additional runoff ($\mathrm{RU_{add}}$).

$$\mathrm{RU_{add}} = \Gamma \times \mathrm{ME_{add}} \qquad (4)$$

Assuming that the residual misfit between reconstructed and observed SMB ($\Delta$SMB) for the different GICs ablation sites is ascribable to underestimated runoff in narrow ablation zones, $\mathrm{RU_{add}}$ is then scaled by a factor $f_{scale}$ (0.682), obtained by computing a least-squares fit minimizing $\Delta$SMB using both GICs and GrIS ablation measurements. The justification for including GrIS observations is that GICs SMB is not independent of the GrIS as the regression parameters were extrapolated from the ice sheet margins onto the GICs. The fact that $f_{scale} < 1$ indicates a slight overestimation of the melt adjustment calculated in equation (3), which could be due to the uncertainties in clouds[39] and/or ice albedo underestimation at the ice caps margins.

$$f_{scale} \frac{\sum \Delta\mathrm{SMB} \times \mathrm{RU_{add}}}{\sum (\mathrm{RU_{add}})^2} \qquad (5)$$

The adjusted amount of runoff ($\mathrm{RU_{v1.0}}$) is obtained by adding $\mathrm{RU_{add}}$ to the downscaled runoff ($\mathrm{RU_{v0.2}}$):

$$\mathrm{RU_{v1.0}} = \mathrm{RU_{v0.2}} + f_{scale} \times \mathrm{RU_{add}} \qquad (6)$$

The corrected melt ($\mathrm{ME_{v1.0}}$) is obtained in a similar manner:

$$\mathrm{ME_{v1.0}} = \mathrm{ME_{v0.2}} + \mathrm{ME_{add}} \qquad (7)$$

Refreezing ($\mathrm{RF_{v1.0}}$) is estimated as a residual between adjusted melt, runoff and rainfall:

$$\mathrm{RF_{v1.0}} = \mathrm{RA} + \mathrm{ME_{v1.0}} - \mathrm{RU_{v1.0}} \qquad (8)$$

The downscaled SMB data set resulting from the combined elevation correction and runoff adjustment is referred to as version v1.0.

**Precipitation correction.** To eliminate the systematic negative SMB bias of RACMO2.3 in Greenland's accumulation zones[16] ($-37.5$ mm w.e. yr$^{-1}$), daily precipitation totals from v0.2 were adjusted in regions presenting a positive annual cumulative SMB in v1.0:

$$\mathrm{PR_{v1.0}} = \mathrm{PR_{v0.2}} \times \left(1 + \frac{\sigma_{SMB}}{\mathrm{PR_{v0.2}^a}}\right) \qquad (9)$$

where $\mathrm{PR_{v1.0}}$ is the daily adjusted total precipitation v1.0, $\mathrm{PR_{v0.2}}$ is the daily bi-linearly interpolated total precipitation v0.2, $\mathrm{PR_{v0.2}^a}$ is the annual cumulative bi-linearly interpolated total precipitation v0.2 and $\sigma_{SMB}$ is the accumulation zone SMB bias in the downscaled product v1.0.

**Product uncertainty.** Assuming that the remaining discrepancies in Fig. 1b consist of a systematic bias due to model uncertainty, combined with additional random scatter, attributed to observations, a product uncertainty can be obtained by integrating the average accumulation and ablation zone biases and adding them as if they were independent. To that end, SMB measurements (Fig. 1b) were binned in 500 mm w.e. bins for which the mean bias was estimated, that is, modelled minus measured SMB. The average uncertainty that results is 247 mm w.e. for the ablation zones (8 bins) and 135 mm w.e. for the accumulation zones (6 bins). The product SMB uncertainty of 15.7 Gt yr$^{-1}$ ($\sim$40%) is obtained by summing the mean ablation bias integrated over the GICs ablation zones (45,600 km$^2$) and the mean accumulation bias over the whole GICs area (81,400 km$^2$), to account for a potential precipitation bias in the ablation zones.

$$\mathrm{Uncertainty} = \sqrt{(\mathrm{bias_{ablation}} \times \mathrm{area_{ablation}})^2 + (\mathrm{bias_{accumulation}} \times \mathrm{area_{GICs}})^2} \qquad (10)$$

Average biases for the GrIS accumulation (22 mm w.e.) and ablation zones (170 mm w.e.) have been calculated in a similar fashion and provided a product SMB uncertainty of 52.5 Gt yr$^{-1}$ ($\sim$20%). These calculations were repeated for 250 mm w.e. SMB bins, which yielded similar results.

**Breakpoint analysis.** We applied the segmented regression method described in ref. 40 to retrieve breakpoints, that is, structural changes, in the GICs-integrated refreezing time series. To detect a break point, the technique fits a regression function, consisting of segments with different slopes, to the studied data set. The algorithm calculates multiple continuous regressions before and after the break points and determines an optimized solution. A confidence interval can be estimated for the resulting break point by using the 95% Wald-based statistics. In this study, we used the segmented regression method to identify a tipping point in the GICs refreezing capacity at 1997 ± 5 years.

**Data availability.** The daily, 1 km SMB data set v1.0 presented in this study is freely available from the authors without conditions. The background in Supplementary Fig. 1 stems from Landsat satellite imagery freely available at https://landsat.usgs.gov/.

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

## Acknowledgements

B.N., W.J.v.d.B., B.W., J.T.M.L. and M.R.v.d.B. acknowledge support from the Polar Programme of the Netherlands Organization for Scientific Research (NWO/ALW) and the Netherlands Earth System Science Centre (NESSC). I.H. and the GIMP project are supported by the U.S. National Aeronautics and Space Administration (NASA). H.M. and M.C. acknowledge support from the Programme for Monitoring of the Greenland Ice Sheet (PROMICE), funded by the Danish Energy Agency's (DANCEA) program. All maps were designed and processed using the free Geographic Information System software QGIS (http://www.qgis.org/en/site/).

## Author contributions

B.N. prepared the manuscript, carried out the RACMO2.3 simulation and produced the downscaled data set at 1 km. B.N., W.J.v.d.B. and M.R.v.d.B. conceived the downscaling procedure and analysed the data. H.M. provided the Greenland ablation data set. I.H. provided the GIMP DEM and grounded ice mask. M.C. provided the PROMICE ice mask. S.L. processed the 1 km MODIS albedo product and performed the breakpoint analysis. G.M. and B.W. produced and analysed the ICESat/CryoSat-2 data sets. J.T.M.L. contributed by preparing mask files and improving figures. All authors commented on the manuscript.

## Additional information

**Competing interests:** The authors declare no competing financial interests.

