## [Peer Review File · Nature Communications]

Reviewers' comments:

Reviewer #1 (Remarks to the Author):

The manuscript entitled "A tipping point in refreezing accelerates mass loss of Greenland's glaciers and ice caps" presents downscaled surface mass balance (SMB) components for the Greenland ice sheet (GrIS) and its peripheral glaciers and ice caps (GICs). Regional climate model output (11-km) is extrapolated onto the GIC domain, and downscaling of SMB components to a 1-km resolution is performed using elevation corrections, runoff and melt adjustments, as well as precipitation corrections. A wide range of observational data is used to optimize the downscaling and to validate the results. The authors discuss trends in surface mass balance components, with a main focus on refreezing, and detect a recent reduction of refreezing for the GICs and a continued buffering role of refreezing for the GrIS. The year 1997 is presented as the 'tipping point' after which loss of firn space reduces the potential for refreezing on the GICs. Spatial differences across Greenland are apparent, with stronger developments in the north than in the south.

The authors have used state-of-the-art methods to generate a valuable 1-km data-set of SMB components. The methods are described in appropriate detail and references are given when needed. The presented results are novel and importantly show a different role of refreezing for the GICs than for the GrIS. This has severe implications for the sensitivity of glacier runoff in the two regions to ongoing changing climate conditions. The manuscript provides a substantial contribution to our understanding of the changing role of refreezing on glacier mass loss in a warming climate. The manuscript is very well written, the presentation of the results is done in a well-structured manner and the figures are clear. I only have minor comments, which are listed below.

Comments:

L39-46 & Fig 1: Here the authors discuss a validation of simulated SMB against measurements for 101 GICs sites. In Figure 1b, the good overall match between modelled and observed data is illustrated in a scatterplot and the five panels in Figure 1a show that the model performs well in different regions. In the next paragraph (L47-55) time-series of modelled area-averaged SMB are compared to ICESat/Cryosat-2 estimates, showing good agreement for the period 2004-2015. However, with the PROMICE data at hand, a longer term (multi-decadal) time-series comparison of modelled and observed mass balance would have been possible. This would yield valuable insight into the model's ability to accurately simulate long-term SMB trends, which is relevant for this work given the focus on multi-decadal developments.

Fig. 3: Sublimation is mentioned as one of the SMB products in the 1-km dataset. In Figure 3, sublimation is not shown, probably because the involved fluxes are very small. Still, it would be helpful to mention this somewhere in the text or figure caption.

L72-90: This paragraph discusses why a tipping point can be found for refreezing on the GICs and not on the GrIS. In the current description the difference of refreezing between the two regions is mainly ascribed to the reduced available pore-space for the GICs, which is less of an issue for the GrIS. I think this discussion needs to be expanded or reformulated somewhat. In general, refreezing is elevation-dependent with a maximum occurring in the lower accumulation zone. High-up in the accumulation zones refreezing may be lower due to limited melt-water supply. Lower down in the ablation zone, refreezing is also smaller but may still be a substantial buffer against mass loss, mainly because refreezing causes the seasonal snow to maintain its mass longer. I think it would be good to link the discussion of the tipping point of refreezing to upward movement of the lower accumulation zones. On the GICs, accumulation zones are shrinking due to firn line retreat, and the elevation where maximum refreezing may occur will in many cases move above the maximum elevation of the glacier or ice cap itself. On the GrIS, the latter effect is less significant and the upward retreat of the firn line simply means an upward shift of the elevation band with maximum refreezing rates. I would like to see some of this discussion in the manuscript.

In order to illustrate the different role of refreezing for the GICs and the GrIS it could be insightful to plot height profiles of mean refreezing for the GICs and the GrIS, comparing the periods 1958-1996 and 1997-2015.

L99-102: It seems a bit surprising to me that the integrated SMB components for Hans Tausen, located in the north, are similar to those of all of Greenland's GICs, because in Fig. S4 you show there is a clear discrepancy in SMB trends between the north and the south.

L104-106: To support this statement, and in addition to the previous suggestion to include height profiles of refreezing, it might also be insightful to show height profiles of runoff for both the GICs and the GrIS for two periods (1958-1996 and 1997-2015). My suggestion would be to replace the current figure 4 with such a figure, while moving the current figure 4 to the supplementary material.

L120: Does 'sublimation (SU)' only include moisture transport from solid to gas or also vice versa, i.e. is deposition or riming also accounted for?

L121: Why is rainfall not corrected for elevation? Since air temperature is elevation-dependent I would expect a similar strong relation between rainfall and elevation.

L139: For the melt adjustment, Eq (3) seems to assume that the additional melt energy equals the additional SW energy absorbed due to the albedo difference. This is however only valid when the surface temperature is already at melting point. When the surface temperature is below melting point additional absorbed solar energy will first be used to heat the surface to melting point, and only the remaining energy will be used to melt the surface. Could this lead to a potential overestimation of calculated melt rates?

L151: Related to the previous comment, could it be that $f_{\text{scale}} (<1)$ is currently compensating for an overestimation of calculated melt rates by using Eq. (3)?

Textual corrections:

L27: With the current formulation it is not clear to what 'this method' refers to. Is it the assumption of negligible solid ice discharge or is it the used methods to generate the 1-km SMB product?

L60: Please replace 'from the GrIS' with 'from the GrIS refreezing regime'.

L63: Please add 'for' after 'compensate'.

L87: I would avoid the term 'irreversible' here, maybe use 'long-term'; instead.

L133, Eq. 2: I would suggest to use a symbol for 'elevation'.

L139: $SW_{\text{direct}}/L_{\text{f}}$ can be taken outside the brackets.

L144: Am I right that ξ is a value close to 1, depending on orientation of the slope?

L151-152: I think 'minimizing the difference between ΔSMB and RU_{add} ' can read 'minimizing ΔSMB '.

L168: Eq (9) can also be written as: $PR_{v1.0} = PR_{v0.2} (1 + \sigma/PR_{v0.2}^a)$

Figure 4: Please replace 'snowfall-to-rainfall ratio' by 'snowfall-to-precipitation ratio'.

Figure S3: The figure appears to be upside-down, at least for the column titles.

Reviewer #2 (Remarks to the Author):

Graham Cogley, September 2016

General Comments

This paper uses downscaled outputs of RACMO2.3, a much-used high-resolution regional climate model, to explore the evolution of the components of mass balance at still finer resolution (1 km) for the peripheral glaciers of Greenland. The main conclusion is that, as summarized in Figure 3c, these glaciers – unlike the ice sheet – have been experiencing progressively less percolation of surface meltwater to depth since about 1997. A natural and convincing interpretation is that the storage capacity of their firn layers (between the surface snow and the ice) has begun to “saturate”: that is, that the void spaces have filled up to the extent that more and more surface meltwater is obliged to run off instead of percolating. Hence the acceleration of mass loss mentioned in the manuscript’s title.

I think I buy the authors’ argument, and the model results on which it is based, but I have a number of concerns. Some are terminological: I know that “refreezing” is standard jargon in ice-sheet modelling for what I have described above without using the word. But if the meltwater does not run off, but percolates instead, it does not matter in strict mass-balance terms whether it actually refreezes. And yet again I have had a hard time reading a manuscript because it adopts the ice-sheet community’s use of total rather than specific (per-unit-area) units – the latter being preferable for purposes such as the present one. On more substantial matters, I am concerned at L52-54 about the authors’ error model, which assumes that random errors are equal to a bias that they have identified (and corrected); and about the complete absence of “ground truth” with which to validate their simulated estimates of “refreezing” from equation 8 (L162). There is probably nothing to be done about the latter concern, other than to acknowledge the difficulty and to explain the equation in somewhat greater detail.

Having said that, if most or all of the comments below can be addressed in some way then I would see this manuscript as a welcome and significant contribution to the literature.

Substantive Comments

P1

L2

The acronym count can be reduced by one by taking note of the fact that ice caps are also glaciers. Simply replace the horrible “GIC” with “glaciers” (or a cognate) wherever it occurs.

L7-8

“to refreeze meltwater”: “refreeze” is probably the wrong verb; it should be “hold”. Your tipping point appeared because the void spaces were filling up, and was not really affected by whether the meltwater froze (which of course depends on the temperature). Following this train of thought, “refreezing capacity” at P2 L11 should probably be “water-holding capacity”, or even “void space” (considering that you are writing for a general audience). Forster et al. 2014 (Extensive liquid meltwater storage in firn within the Greenland ice sheet, *Nature Geoscience*, **7**(2), 95-98) is pertinent in this context.

P2

L20

I am not sure why ref.13 is cited here, or indeed why it is cited at all. It has nothing to say about mass-balance observations in Greenland. It may be a mistake for Cogley, J.G., 2009, Geodetic and direct mass-balance measurements: comparison and joint analysis, *Annals of Glaciology*, **50**(50), 96-100 – but that says nothing about observations in Greenland either. [See my comment on Table S3 below.] [Note also that it is essential to include the issue number in references to *Annals of Glaciology*; your wrong reference is to a paper published in volume **50** issue 53.]

P3

- L30 How do you treat ref.23's CL2 glaciers ("strongly" connected to the ice sheet without being a part of it)? At L54 it sounds as though you count them as part of the ice sheet, but you need to be explicit.
- L31 What does "using elevation dependence" mean? As a minimum you could say "(as explained in the Methods section)", but it would be better to clarify by rearranging, e.g.: "... RACMO2.3, using regressions against elevation, evaluated at the 11-km resolution of the model, to downscale to the 1-km version of the topography ...".
- L49-50 Why do you plot the cumulative mass balance and not the mass-balance rate? I cannot see the "large difference" you mention between 2012 and 2013, and I would like to know more about why your simulation in dashed blue is so much smoother than Cryosat-2.
- P4
- L50-52 This sentence is fallacious. The similarity with altimetry shows that thinning and mass loss go together, not that ice discharge is negligible. Unless you can restore the logic, this sentence could be deleted without loss. I do not know whether the neglect of ice discharge from peripheral glaciers is something to be worried about; perhaps not, but on the other hand Pfeffer et al. (2014, *Journal of Glaciology*, **60**(221), 537-552; their Table 2) report that of the ~90,000 km² of peripheral ice ~31,000 km², about a third, drains to tidewater.
- L52-54 This error "model" is defective. There is no reason why a bias should be a good estimator of a random error, and your much larger RMSE would be a better starting point. The thing to do with a bias is to correct it (as explained on P9). (The integration over 81,000 km² is not relevant, serving simply to convert to a loss rate of total mass from a rate per unit area.)
- L54-55 "due to the omission of small glaciers (<1 km²) in the original GIMP DEM [or should it be 'the PROMISE ice mask'?]". 8% is a fairly significant underestimate of the extent of peripheral glaciers. You could do a better job by turning to the Randolph Glacier Inventory (<http://www.glims.org/RGI/index.html>) and chopping up all ~19,000 glacier outlines into their 1x1 km cells, but this would probably be an unreasonable amount of work for the present submission. (I cannot guess whether the omitted tiny glaciers are more likely or less likely than the others to experience refreezing.)
- L56 Figure 3: I have never understood why ice-sheet specialists insist on presenting mass changes in total units (Gt a⁻¹; perfect for adding up all the contributions to sea-level change) rather than specific units (mm w.e. a⁻¹; essential for comparing the intensity of processes between ice bodies of different size). Crudely, the ice-sheet refreezing is of the order of 194 mm w.e. a⁻¹ (350 Gt a⁻¹) while the glaciers experience 279 mm w.e. a⁻¹ (25 Gt a⁻¹). These numbers would fit nicely and informatively on a common vertical scale in specific units for Figure 3c.
- L61 It would be a good idea to define "firn" for a general audience. I understand it to be frozen water between the current year's snow and the horizon at which the bubble-closeoff density, about 830 kg m⁻³, is reached.
- L61-62 I suggest changing "increased melt" to "percolation and storage". And "enhanced refreezing" is wrong – there may indeed have been an increase in melting, but the point is that there is now less room for meltwater at depth, no matter how much is produced at the surface; change to "continued percolation".
- L65ff. As with Figure 3, it may be just me (although I do not really think so), but I think the rest of the text would be a lot easier to follow if most of the mass changes were in specific units.
- P5
- L74 Again, ref. 13 is not appropriate.
- L89-90 Scaling up the glaciers' acceleration of loss (by multiplying by the ratio of ice sheet area to glacier area) is yet more absurd than dividing the ice-sheet numbers by 10 in Figure 3c. Make this comparison in specific units.
- P9
- L161-162 Do you know of *any ground truth at all* about "refreezing"? It is essentially impossible to measure it in-situ, but perhaps there are hints to be gained from analyses of the seasonal

evolution of density (and temperature?) profiles, the prevalence of ice lenses in shallow cores, etc. Notwithstanding your rather successful comparisons to observed mass balance, the fact remains that Eq8 is the end of a long chain of educated guesses. As a minimum it needs more discussion. Why is all rainfall assumed to refreeze? Nearer to the start of the chain, how does RACMO2.3 calculate melt ME and runoff RU?

Stylistic Comments

P1

L1 Delete “contiguous”. Same at L17, L27-28 and so on. It is unnecessary anyway, but is also ambiguous because nearly a quarter of the peripheral-glacier area (ref. 23’s CL1 glaciers) is “contiguous” with the ice sheet.

L9

Surely “pre-1997” should be “post-1997”?

P2

L15

Change “glaciated regions” to “glacierized area”.

L26

Why is “changes in” in parentheses? I suspect that it was marked for deletion but not deleted.

P3

L39

“entire Greenland” is not idiomatic English. You have to say “all of” or “the whole of”.

L42

“w.e.” is the preferred abbreviation for “water equivalent”, and you should not omit the essential “a⁻¹” – either here or in later instances of the unit.

L46

Change “later in the main text” to “below”.

L47

Insert “observations” before “over”.

L48

What does “(v1.0)” refer to?

P4

L64

Unfortunately you cannot translate German *Haushalt* as “household”, which is English for a house and its contents (including its human occupants). You have to say “balance” or “budget”. (I am afraid I do not know the Dutch for the German word.)

L68

“for maintaining”.

P5

L81-82

“Tables S1 and S2”; but follow journal style about abbreviating “Table” as at L82. “responses”.

L86-87

Soften “eventually leading to” to “implying eventual”. Change “refreezing” to something like “meltwater retention”.

P6

L91-93

Change “Fig. S4 contrasts” to “we examined”, and cite Figure S4 at the end of the sentence.

L97

“twice as large ... as for southern”.

L105

“zones”.

L110

Change “deteriorated” to “reduced”, and preferably “refreezing” to “meltwater retention [or ‘storage’]”.

P7

L120

“correlation with”.

L124-125

“for the dependence of modelled SMB components on the 11-km”.

L126

Change “glaciated” to “glacierized”.

L126-127

The current pixel is not adjacent to itself, and has already made its appearance a few words earlier. So change “6” to “five” and delete “including the current one”.

L130

Spell out “three”.

L139

The right-hand side of Eq3 would be more readable if it ran “... 0.5 (1+ ξ) $SW_{\text{direct 1-km}}/L_f$ ”.

L141

“(–)” slowed me down by several seconds; expand it to “(dimensionless)”, or just leave it out.

L144

Italicize L_f . “ ξ is the dimensionless ...”. Could you add a short explanation of why ξ is needed, and where you get the tilt (slope?) from?

L146

Delete the unnecessary “specific”, and perhaps change “fraction” to “ratio”. You could then change the period to a colon and delete L147. Italicize Γ at L146,148.

P9
L150 “ascribable”.
L151 Italicize f here and in Eqs 5,6. “least-squares”.
L152 Change “reason” to “justification”.
L166 “daily precipitation totals from v0.2 were”.
P10
L177 You cannot perform a method. Say “we used”.
P12
L215 z..??
L219 Do not capitalize the paper title.
P15
L262 “by preparing mask files and improving figures”.

Fig1 L4: “five”.
L5: “gradients are” should be “is” (there are no fitted lines). “downscaled simulations (b_{mod} ; blue dots) and in-situ data(b_{obs} ; red dots)”.
Panel b: explain b_0 and b_1 . Oddly, b_0 seems to be the slope and b_1 the intercept of the regression line.

Fig3 L4: I think “integrated annual mean” should be “annual”. “integrated” is redundant, and “mean” suggests that you divided by the number of days or something, when actually you just summed over the year.

Fig4 Delete “mean” at L3, but not at L6.

Supplementary Information

P2
L8 “glacierized”.
L17 What is the “original” product? The GIMP DEM? If so, you have to distinguish between the averaging of elevations and whatever was done (possibly nothing) to assign “glacier” codes in the 1-km mask.
L18 Delete “contiguous”.
L19 As in the main text, you should say explicitly here that the CL2 glaciers are treated as part of the ice sheet.
L27 “albedos”.
P3
L39 “grid cell”.
L40 Delete the redundant “adjacent”.
TableS1,S2 Use the same symbols for balance components as in Eq1 of the main text.
TableS3 The references should probably carry the same numbers as in the main text, but once again I am baffled by the mention of Cogley 2009 (see comment above at P2 L20). Here, however, we have confirmation that you are actually citing Marzeion et al. 2012, ref. 15; Cogley 2009 (as corrected above) was the source for in-situ observations in ref.15, but only as a paper documenting a dataset that was updated for ref. 15.
L4: “Uncertainties ... are obtained”. But see comment on the main text (P4 L54).

FigS1 In panels b and c the dazzling red of the ice sheet tends to obscure the information you are trying to convey; perhaps light grey instead as in FigS4?

FigS2 “by regression against elevation”.
FigS4 L2: delete “mean”. L3: “The changes ... are”.

Below, our responses to the individual reviewers' comments are displayed in blue and modifications in the manuscript in orange to facilitate readability.

Reviewer #1 (Remarks to the Author):

The manuscript entitled "A tipping point in refreezing accelerates mass loss of Greenland's glaciers and ice caps" presents downscaled surface mass balance (SMB) components for the Greenland ice sheet (GrIS) and its peripheral glaciers and ice caps (GICs). Regional climate model output (11-km) is extrapolated onto the GIC domain, and downscaling of SMB components to a 1-km resolution is performed using elevation corrections, runoff and melt adjustments, as well as precipitation corrections. A wide range of observational data is used to optimize the downscaling and to validate the results. The authors discuss trends in surface mass balance components, with a main focus on refreezing, and detect a recent reduction of refreezing for the GICs and a continued buffering role of refreezing for the GrIS. The year 1997 is presented as the 'tipping point' after which loss of firn space reduces the potential for refreezing on the GICs. Spatial differences across Greenland are apparent, with stronger developments in the north than in the south.

The authors have used state-of-the-art methods to generate a valuable 1-km data-set of SMB components. The methods are described in appropriate detail and references are given when needed. The presented results are novel and importantly show a different role of refreezing for the GICs than for the GrIS. This has severe implications for the sensitivity of glacier runoff in the two regions to ongoing changing climate conditions. The manuscript provides a substantial contribution to our understanding of the changing role of refreezing on glacier mass loss in a warming climate. The manuscript is very well written, the presentation of the results is done in a well-structured manner and the figures are clear. I only have minor comments, which are listed below.

Comments:

L39-46 & Fig 1: Here the authors discuss a validation of simulated SMB against measurements for 101 GICs sites. In Figure 1b, the good overall match between modelled and observed data is illustrated in a scatterplot and the five panels in Figure 1a show that the model performs well in different regions. In the next paragraph (**L47-55**) time-series of modelled area-averaged SMB are compared to ICESat/Cryosat-2 estimates, showing good agreement for the period 2004-2015. However, with the PROMICE data at hand, a longer term (multi-decadal) time-series comparison of modelled and observed mass balance would have been possible. This would yield valuable insight into the model's ability to accurately simulate long-term SMB trends, which is relevant for this work given the focus on multi-decadal developments.

The PROMICE data compiles measurements going back to the 1890s. In our study we focus on 1958-2015, the period covered by ERA-40 (1958-1978) and ERA-Interim (1979-2015). For validation of our downscaled SMB product, only measurements for the overlapping period can be used. We deem that the period 1958-2015 (58 years) is sufficiently long to estimate significant trends.

Fig. 3: Sublimation is mentioned as one of the SMB products in the 1-km dataset. In Figure 3, sublimation is not shown, probably because the involved fluxes are very small. Still, it would be helpful to mention this somewhere in the text or figure caption.

Thank you for pointing that out. We inserted the following lines in the caption: “Total sublimation (SU) and snow drift erosion (ER) are not included in the above time series as they contribute relatively little to SMB and trends are very small compared to the other components.”

L72-90: This paragraph discusses why a tipping point can be found for refreezing on the GICs and not on the GrIS. In the current description the difference of refreezing between the two regions is mainly ascribed to the reduced available pore-space for the GICs, which is less of an issue for the GrIS. I think this discussion needs to be expanded or reformulated somewhat. In general, refreezing is elevation-dependent with a maximum occurring in the lower accumulation zone. High-up in the accumulation zones refreezing may be lower due to limited melt-water supply. Lower down in the ablation zone, refreezing is also smaller but may still be a substantial buffer against mass loss, mainly because refreezing causes the seasonal snow to maintain its mass longer. I think it would be good to link the discussion of the tipping point of refreezing to upward movement of the lower accumulation zones. On the GICs, accumulation zones are shrinking due to firn line retreat, and the elevation where maximum refreezing may occur will in many cases move above the maximum elevation of the glacier or ice cap itself. On the GrIS, the latter effect is less significant and the upward retreat of the firn line simply means an upward shift of the elevation band with maximum refreezing rates. I would like to see some of this discussion in the manuscript. In order to illustrate the different role of refreezing for the GICs and the GrIS it could be insightful to plot height profiles of mean refreezing for the GICs and the GrIS, comparing the periods 1958-1996 and 1997-2015.

As suggested, we included a new figure (S4, see below) in the Supplementary Information and reformulated **L85-90** in the main manuscript as follow: “Supplementary Fig. 4 confirms these findings by comparing vertical profiles of surface mass fluxes integrated over GICs and GrIS elevation bins, scaled by the maximum height per region (h_{max}), prior to and after 1997. Supplementary Figs. 4a and b show that the equilibrium line ($SMB = 0$) of the GICs moved significantly upward, i.e. from 0.61 to 0.71 of h_{max} , and is now situated well above the peak in the hypsometry (0.62, Fig. 4d). In combination with decades of increased melt, which depleted firn pore space, the GICs firn layer is no longer capable of buffering the excess meltwater production. As a result, runoff increases at the same rate as melt and fully governs the GICs mass loss (Supplementary Fig. 4c). Supplementary Figs. 4a and b also show that rainfall is a small (6%) fraction of the liquid water flux available at the firn layer top, which is dominated by melt. For the GrIS the equilibrium line has moved upwards from 0.33 to 0.40 of h_{max} (Supplementary Figs. 4e and f), but remains well below the maximum in the GrIS hypsometry (0.73, Supplementary Fig. 4h). Therefore, a significant part of the excess melt is buffered by refreezing (Supplementary Fig. 4g) and runoff remains constant above 0.61 of h_{max} . Although formation of ice lenses may reduce the retention efficiency in the lower accumulation zone^{27,28}, we conclude that the extensive and elevated inland firn

area of the GrIS (Supplementary Fig. 4h) maintains its refreezing capacity for now. As a result, the acceleration of GrIS surface mass loss is less than half that of the GICs (-6.1 ± 2.4 mmw.e. yr⁻² vs. -13.5 ± 7.4 mmw.e. yr⁻².”

Figure S4: Shrinking of the accumulation zones: Vertical profiles of surface mass fluxes integrated over GICs (upper row) and GrIS (lower row) elevation bins, scaled by the maximum height per region (h_{max}), for the period 1958-1996 (a and e), 1997-2015 (b and f) and the difference between the two periods (1997-2015 minus 1958-1996, c and g). SMB components are spatially integrated within normalized elevation bins (h/h_{max}) of magnitude 0.05. For the GICs, SMB components are first integrated in elevation bins over twelve individual regions, each with different h_{max} (boxes in Supplementary Fig. 5); the GICs-integrated SMB components are obtained by summing the contribution of the twelve regions to each scaled elevation bin. Supplementary Figs. 4 d) and h) show the scaled hypsometries, i.e. total area occupied by each elevation bin, for the GICs and the GrIS, respectively.

L99-102: It seems a bit surprising to me that the integrated SMB components for Hans Tausen, located in the north, are similar to those of all of Greenland's GICs, because in Supplementary Fig. 4 you show there is a clear discrepancy in SMB trends between the north and the south.

This sentence was confusing and we reformulated as: "The Hans Tausen region shows a small steady mass loss [...] afterwards (Supplementary Fig. 6)." We also included the Hans Tausen time series as Supplementary Figure 6 in the

Supplementary Information (see below).

Figure S6: Mass flux evolution in the Hans Tausen region. Time series of annual cumulative SMB components over the Hans Tausen ice cap and surroundings (Black box 5 in Fig. 1) for the period 1958-2015. Trends in SMB components are represented as coloured dashed lines (1997-2015).

L104-106: To support this statement, and in addition to the previous suggestion to include height profiles of refreezing, it might also be insightful to show height profiles of runoff for both the GICs and the GrIS for two periods (1958-1996 and 1997-2015). My suggestion would be to replace the current figure 4 with such a figure, while moving the current figure 4 to the supplementary material.

See our response to the question at **L72-90**. We decided to include the additional figure in the Supplementary Information, but are willing to swap Fig. 4 with Supplementary Figure 4 at the discretion of the editor.

L120: Does 'sublimation (SU)' only include moisture transport from solid to gas or also vice versa, i.e. is deposition or riming also accounted for?

Yes, periods of deposition are included in the averages.

L121: Why is rainfall not corrected for elevation? Since air temperature is elevation-dependent I would expect a similar strong relation between rainfall and elevation. Indeed, rainfall correlates well with elevation (see Figure below) and can easily be downscaled; this was not done because it is a sub-component of the SMB. In RACMO2.3, precipitation is modeled by solving the atmosphere dynamics and physics at different levels and involves many additional components, i.e. specific humidity, wind speed and direction, air temperature and near surface temperature (or topography). Using the elevation regressions would only account for (changes in) near surface temperature without considering the availability of moisture in the atmosphere or the wind circulation patterns, which are both essential to accurately time and distribute solid and liquid precipitation. By downscaling rainfall, we might slightly improve the rainfall-to-snowfall ratio in low-lying regions but this would not affect much the downscaled SMB as rainfall remains negligible compared to snowfall accumulation.

In addition, the contribution of rain to refreezing is implicitly included in our analysis but not explicitly discussed in the text. To remedy this, we have now included the vertical profile of rainfall flux in Supplementary Fig. 4; it shows that the rainfall flux (change) is small compared to melt flux (change), and occurs mainly in the ablation zone (SMB < 0) where the impact on refreezing is small. To clarify this, we added the following sentence to the main text: “Supplementary Fig. 4a and b also show that rainfall is a small (6%) fraction of the liquid water flux available at the firn layer top, which is dominated by melt.”

Ancillary Figure: Correlation to elevation of annual mean a) rainfall modeled by RACMO2.3 and calculated on the 11-km grid for the period 1958-2015.

L139: For the melt adjustment, Eq (3) seems to assume that the additional melt energy equals the additional SW energy absorbed due to the albedo difference. This is however only valid when the surface temperature is already at melting point. When the surface temperature is below melting point additional absorbed solar energy will first be used to heat the surface to melting point, and only the remaining energy will be used to melt the surface. Could this lead to a potential overestimation of calculated melt rates?

No, because melt adjustment is only applied when both runoff and melt are nonzero at the surface. This means that surface temperatures in RACMO2.3 are at the melting point. However, overestimation of melt rates could still occur due to uncertainties in clouds and ice albedo representation close to the ice caps margins. To highlight this, we inserted the following sentence at **L154**: “The fact that $f_{scale} < 1$ indicates a slight overestimation of the melt adjustment calculated in Eq. (3), which could be due to uncertainties in clouds (Van Tricht et al., 2016) and/or ice albedo underestimation at the ice caps margins.”

L151: Related to the previous comment, could it be that $f_{scale} (<1)$ is currently compensating for an overestimation of calculated melt rates by using Eq. (3)?
See our response to the previous question.

Textual corrections:

L27: With the current formulation it is not clear to what ‘this method’ refers to. Is it the assumption of negligible solid ice discharge or is it the used methods to generate the 1-km SMB product?

I meant the “downscaling method”. This is now clarified in the revised manuscript.

L60: Please replace ‘from the GrIS’ with ‘from the GrIS refreezing regime’. OK.

L63: Please add ‘for’ after ‘compensate’. OK.

L87: I would avoid the term ‘irreversible’ here, maybe use ‘long-term’; instead. OK.

L133, Eq. 2: I would suggest to use a symbol for ‘elevation’.

We reformulated the equation as: “ $X_{v0.2} = a_{1km} + b_{1km} \times h_{1km}$ ”

L139: SW_{direct}/L_f can be taken outside the brackets. We reformulated accordingly.

L144: Am I right that ξ is a value close to 1, depending on orientation of the slope? Depending on the orientation and the slope of the 1 km grid cells, ξ ranges from 0 to 2 at most. For instance, $\xi = 0$ for a grid cell oriented to the north, while $\xi > 1$ for grid cells oriented to the south. $\xi = 1$ for horizontal pixels in the 1 km grid. To clarify this, we inserted the following lines: “To account for the slope of the glacier surface, the dimensionless correction factor for a tilted plane ξ is applied to the direct component of modeled downward shortwave radiation. This correction is required as RACMO2.3 models radiation assuming a horizontal surface. This factor ranges from 0 for north sloping glaciers to values larger than 1 for south oriented slopes.”

L151-152: I think 'minimizing the difference between ΔSMB and RU_{add} ' can read 'minimizing ΔSMB '. OK.

L168: Eq (9) can also be written as: $\text{PR}_{v1.0} = \text{PR}_{v0.2} (1 + \sigma/\text{PR}_{v0.2}^a)$

We reformulated accordingly.

Figure 4: Please replace 'snowfall-to-rainfall ratio' by 'snowfall-to-precipitation ratio'. OK.

Figure S3: The figure appears to be upside-down, at least for the column titles.

Figure S3 should be read horizontally, which solves the problem.

Reviewer #2: Graham Cogley, September 2016

General Comments:

This paper uses downscaled outputs of RACMO2.3, a much-used high-resolution regional climate model, to explore the evolution of the components of mass balance at still finer resolution (1 km) for the peripheral glaciers of Greenland. The main conclusion is that, as summarized in Figure 3c, these glaciers – unlike the ice sheet – have been experiencing progressively less percolation of surface meltwater to depth since about 1997. A natural and convincing interpretation is that the storage capacity of their firn layers (between the surface snow and the ice) has begun to “saturate”: that is, that the void spaces have filled up to the extent that more and more surface meltwater is obliged to run off instead of percolating. Hence the acceleration of mass loss mentioned in the manuscript’s title. I think I buy the authors’ argument, and the model results on which it is based, but I have a number of concerns. Some are terminological: I know that “refreezing” is standard jargon in ice-sheet modelling for what I have described above without using the word. But if the meltwater does not run off, but percolates instead, it does not matter in strict mass-balance terms whether it actually refreezes. And yet again I have had a hard time reading a manuscript because it adopts the ice-sheet community’s use of total rather than specific (per-unit-area) units – the latter being preferable for purposes such as the present one. On more substantial matters, I am concerned at L52-54 about the authors’ error model, which assumes that random errors are equal to a bias that they have identified (and corrected); and about the complete absence of “ground truth” with which to validate their simulated estimates of “refreezing” from equation 8 (L162). There is probably nothing to be done about the latter concern, other than to acknowledge the difficulty and to explain the equation in somewhat greater detail. Having said that, if most or all of the comments below can be addressed in some way then I would see this manuscript as a welcome and significant contribution to the literature.

Substantive Comments:

P1

L2: The acronym count can be reduced by one by taking note of the fact that ice caps are also glaciers. Simply replace the horrible “GIC” with “glaciers” (or a cognate) wherever it occurs.

While we agree that ice caps are also glaciers, so are ice sheets! This means that the distinction would be gone if we were to use “glaciers” instead of GIC. That is why we prefer to keep the acronym “GIC” in the revised manuscript; it is also commonly used in the scientific literature (e.g. Rastner et al., 2011; Citterio et al., 2013; Machguth et al. 2013).

L7-8: “to refreeze meltwater”: “refreeze” is probably the wrong verb; it should be “hold”. Your tipping point appeared because the void spaces were filling up, and was

not really affected by whether the meltwater froze (which of course depends on the temperature).

In the absence of perennial firn aquifers, retention of meltwater in the firn purely consists of meltwater refreezing. According to Kuipers Munneke et al. (2014), there are no or only very localized perennial firn aquifers on GICs (see Fig. 4c in Kuipers Munneke et al. (2014)). In view of this, we decided to keep the verb “refreeze” in the revised manuscript, which is also commonly used in literature.

- **Kuipers Munneke, P., S. R. M. Ligtenberg, M. R. van den Broeke, J. H. van Angelen and R. R. Forster. 2014. Explaining the presence of perennial liquid water bodies in the firn of the Greenland Ice Sheet. *Geophys. Res. Lett.*, doi:10.1002/2013GL058389.**

P2

L11: Following this train of thought, “refreezing capacity” should probably be “water-holding capacity”, or even “void space” (considering that you are writing for a general audience). **Forster et al. 2014** (Extensive liquid meltwater storage in firn within the Greenland ice sheet, *Nature Geoscience*, 7(2), 95-98) is pertinent in this context.

Please see the previous comment: in the absence of perennial firn aquifers, the retention of meltwater in the firn layer of the GICs is caused purely by refreezing. Therefore, we decided to keep the phrase “refreezing capacity”.

L20: I am not sure why ref.13 is cited here, or indeed why it is cited at all. It has nothing to say about mass-balance observations in Greenland. It may be a mistake for Cogley, J.G., 2009, Geodetic and direct mass-balance measurements: comparison and joint analysis, *Annals of Glaciology*, 50(50), 96-100 – but that says nothing about observations in Greenland either. [See my comment on Table S3 below.] [Note also that it is essential to include the issue number in references to *Annals of Glaciology*; your wrong reference is to a paper published in volume 50 issue 53.]

We agree and removed the reference accordingly.

P3

L30: How do you treat ref.23’s CL2 glaciers (“strongly” connected to the ice sheet without being a part of it)? At L54 it sounds as though you count them as part of the ice sheet, but you need to be explicit.

We clarified this by inserting “[...] ice sheet, including glaciers strongly connected to the ice sheet (corresponding to connectivity level CL2 in Ref.23), [...]” at **L29**.

L31: What does “using elevation dependence” mean? As a minimum you could say “(as explained in the Methods section)”, but it would be better to clarify by rearranging, e.g.: “... RACMO2.3, using regressions against elevation, evaluated at the 11-km resolution of the model, to downscale to the 1-km version of the topography ...”.

We reformulated as follow: “[...] model RACMO2.3 using regressions of SMB components against elevation estimated at the model resolution of 11 km. These regressions are then applied to a down-sampled 1-km version ...”.

L49-50: Why do you plot the cumulative mass balance and not the mass-balance rate? I cannot see the “large difference” you mention between 2012 and 2013, and I would like to know more about why your simulation in dashed blue is so much smoother than Cryosat-2.

ICESat and Cryosat-2 measure the temporal evolution of the GICs volume and mass, i.e. their cumulative volume and mass change in time. To plot the mass balance rate, we would need to take the difference between successive measurements (temporal derivative), which would induce substantial noise in the time series due to errors in the observations. Plotting the volume/mass anomaly as a function of time is the standard approach for this type of measurements (e.g., Gardner et al., 2011; Khvorostovsky et al., 2012; Flament et al., 2014; McMillan et al., 2016).

- T. Flament, F. Rémy, Dynamic thinning of Antarctic glaciers from along-track repeat radar altimetry. *J. Glaciol.* 58, 830–840 (2012). doi:10.3189/2012JoG11J118
- S. Gardner, G. Moholdt, B. Wouters, G. J. Wolken, D. O. Burgess, M. J. Sharp, J. G. Cogley, C. Braun, C. Labine, Sharply increased mass loss from glaciers and ice caps in the Canadian Arctic Archipelago. *Nature* 473, 357–360 (2011). doi:10.1038/nature10089
- Khvorostovsky K. Merging and Analysis of Elevation Time Series Over Greenland Ice Sheet From Satellite Radar Altimetry. *IEEE Transactions on Geoscience and Remote Sensing.* 2012;50(1).
- McMillan, M., et al. (2016), A high-resolution record of Greenland mass balance, *Geophys. Res. Lett.*, 43, 7002–7010, doi:10.1002/2016GL069666.

P4

L50-52: This sentence is fallacious. The similarity with altimetry shows that thinning and mass loss go together, not that ice discharge is negligible. Unless you can restore the logic, this sentence could be deleted without loss. I do not know whether the neglect of ice discharge from peripheral glaciers is something to be worried about; perhaps not, but on the other hand Pfeffer et al. (2014, *Journal of Glaciology*, 60(221), 537-552; their Table 2) report that of the ~90,000 km² of peripheral ice ~31,000 km², about a third, drains to tidewater.

We removed the sentence accordingly.

L52-54: This error “model” is defective. There is no reason why a bias should be a good estimator of a random error, and your much larger RMSE would be a better starting point. The thing to do with a bias is to correct it (as explained on P9). (The integration over 81,000 km² is not relevant, serving simply to convert to a loss rate of total mass from a rate per unit area.)

Using the RMSE as estimator would overestimate the spatially integrated uncertainty. The downscaled 1 km SMB product uncertainty includes both random

errors, originating from both the observations and the model, and systematic errors, mainly originating from the model. Since we integrate over a large area and many glaciers, random errors will tend to cancel out, leaving the systematic error for the error estimate. So, assuming that the available observations provide a representative assessment of the quality of the 1 km product for all GICs, our bias correction largely removes the error of the model estimate.

Upon giving this a fair amount of thought, we agree that simply integrating the mean bias might provide a too optimistic estimate of the product uncertainty. For example, certain SMB-zones might be over- or underrepresented in the observational dataset. To address this issue, we estimated the bias for individual 500 mmw.e. bins. We then averaged these absolute biases for the ablation zone (8 bins; $bias_{ablation}$) and for the accumulation zone (2 bins; $bias_{accumulation}$) separately. Next, we calculated the total uncertainty of model SMB as the (assumed independent) uncertainties in accumulation and ablation, as follows:

$$Uncertainty_{model} = \sqrt{(bias_{ablation} \times area_{ablation})^2 + (bias_{accumulation} \times area_{GICs})^2}$$

Note that we integrated the accumulation bias over the full area of GICs since the ablation zone is also affected by biases in solid precipitation. This provides an uncertainty of 15.7 Gt/yr (~40%) in SMB for the GICs. A similar calculation provides a model uncertainty of 52.5 Gt/yr (~20%) for the SMB of the GrIS. We also repeated the calculations using 250 mmw.e. bins and obtained similar results: 15.9 Gt/yr for the GICs and 50.8 Gt/yr for the GrIS, indicating that this method is robust for bin size.

L52-54 as follow: “The uncertainty in downscaled SMB was estimated at 15.7 Gt/yr (~40%, see Methods). Note that the Greenland’s GICs area of ~81,400km² used in this study is smaller by ~8% than previous [...]” We also included a new Section in the methods: “**Product uncertainty:** Assuming that the remaining discrepancies in Fig. 1b consist of a systematic bias due to model uncertainty, combined with additional random scatter, attributed to observations, a product uncertainty can be obtained by integrating the average accumulation and ablation zones biases and adding them as if they were independent. To that end, SMB measurements (Fig. 1b) were binned in 500 mmw.e. bins for which the mean bias was estimated, i.e. modeled minus measured SMB. The average uncertainty that results is 247 mmw.e for the ablation zones (8 bins) and at 135 mmw.e for the accumulation zones (6 bins). The product SMB uncertainty of 15.7 Gt/yr (~40%) is obtained by summing the mean ablation bias integrated over the GICs ablation zones (45,600 km²) and the mean accumulation bias over the whole GICs area (81,400 km²), to account for potential precipitation bias in the ablation zones.

$$Uncertainty_{model} = \sqrt{(bias_{ablation} \times area_{ablation})^2 + (bias_{accumulation} \times area_{GICs})^2}$$

Average biases for the GrIS accumulation (22 mmw.e.) and ablation zones (170 mmw.e.) have been calculated in a similar fashion and provided a product SMB uncertainty of 52.5 Gt/yr (~20%). These calculations were repeated for 250 mmw.e. SMB bins, which yielded similar results.”

L54-55a: “due to the omission of small glaciers (<1 km²) in the original GIMP DEM [or should it be ‘the PROMISE ice mask?’]”. We reformulated accordingly.

L54-55b: 8% is a fairly significant underestimate of the extent of peripheral glaciers. You could do a better job by turning to the Randolph Glacier Inventory (<http://www.glims.org/RGI/index.html>) and chopping up all ~19,000 glacier outlines into their 1x1 km cells, but this would probably be an unreasonable amount of work for the present submission. (I cannot guess whether the omitted tiny glaciers are more likely or less likely than the others to experience refreezing.)

For consistency, we decided to use the GIMP DEM to represent both the topography and the ice mask at 1 km. An 8% difference is certainly significant, however, the ice bodies omitted by the GIMP DEM are small compared to the main ice caps and the real GICs surface area remains elusive; the figure below shows that while the GIMP DEM (blue and green) misses some small ice bodies, it also resolves additional small ice masses not covered by the Randolph Glacier Inventory (RGI; yellow).

Ancillary Figure: Promise mask in the northwestern corner of the center east Greenland region (uppermost right black box in Supplementary Fig. 5). The ice sheet and strongly connected ice bodies (CL2) are displayed in red, the detached and attached glaciers and ice caps are in blue and green, respectively. The Randolph Glacier Inventory (RGI) is displayed in yellow.

L56 Figure 3: I have never understood why ice-sheet specialists insist on presenting mass changes in total units (Gt a⁻¹; perfect for adding up all the contributions to sea-level change) rather than specific units (mm w.e. a⁻¹; essential for comparing the intensity of processes between ice bodies of different size). Crudely, the ice-

sheet refreezing is of the order of 194 mm w.e. a⁻¹ (350 Gt a⁻¹) while the glaciers experience 279 mm w.e. a⁻¹ (25 Gt a⁻¹). These numbers would fit nicely and informatively on a common vertical scale in specific units for Figure 3c.

To accommodate this request, we added a second y-axis on Figure 3c. However, this resulted in a confusing picture because absolute numbers are scaled for GrIS to accommodate both lines in a single graph. We decided to keep the units in Gt yr⁻¹ and we included numbers in specific units in Tables S1 and S2.

L61: It would be a good idea to define “firn” for a general audience. I understand it to be frozen water between the current year’s snow and the horizon at which the bubble-closeoff density, about 830 kg m⁻³, is reached.

To make this clear, we inserted: “[...] deteriorating firn layer, the porous, multiyear snow layer between surface fresh snow (~350 kg m⁻³) and the underlying ice (~900 kg m⁻³). Decades of ...”.

L61-62: I suggest changing “increased melt” to “percolation and storage”. And “enhanced refreezing” is wrong – there may indeed have been an increase in melting, but the point is that there is now less room for meltwater at depth, no matter how much is produced at the surface; change to “continued percolation”.

Based our response to comment at **L7-8** and **L11**, we decided to keep these sentences unchanged.

L65ff: As with Figure 3, it may be just me (although I do not really think so), but I think the rest of the text would be a lot easier to follow if most of the mass changes were in specific units.

See our answer to comment **L56 Figure 3**; in order not to confuse the reader, we prefer to stick with a single unit Gt yr⁻¹ in the main text. We included numbers in mmw.e. yr⁻¹ in Tables S1 and S2.

P5

L74: Again, ref. 13 is not appropriate.

We removed the reference accordingly.

L89-90: Scaling up the glaciers’ acceleration of loss (by multiplying by the ratio of ice sheet area to glacier area) is yet more absurd than dividing the ice-sheet numbers by 10 in Figure 3c. Make this comparison in specific units.

We calculated the mass loss acceleration in mmw.e./yr² and reformulated as follow: “As a result, the acceleration of GrIS surface mass loss is less than half that of the GICs (-6.1 ± 2.4 mmw.e. yr⁻² vs. -13.5 ± 7.4 mmw.e. yr⁻²).”

P9

L161-162: Do you know of any ground truth at all about “refreezing”? It is essentially impossible to measure it in-situ, but perhaps there are hints to be gained from analyses of the seasonal evolution of density (and temperature?) profiles, the prevalence of ice lenses in shallow cores, etc.

RACMO2 has been thoroughly evaluated against in situ SMB measurements (Ettema et al., 2010; Van Angelen et al. 2012 and Noël et al. 2015) and against mass changes

from GRACE gravity data (Van den Broeke et al., 2009,2016; Van Angelen et al., 2013). The 1 km GrIS SMB product also shows good agreement with in situ SMB measurements in the ablation (governed by runoff) and the accumulation zones (governed by precipitation; Noël et al., 2016). Furthermore, comparisons between modelled and measured surface energy budget components also show relatively good agreement (Ettema et al., 2010; Noël et al., 2015), indicating that surface melt is well represented in RACMO2. We conclude that RACMO2, and the downscaled product, succeed at modelling the spatial and temporal variability of the GrIS main SMB components (runoff, melt and precipitation). Similar conclusions can be drawn for the GICs as shown from in situ SMB measurements in Figs. 1 a-b and altimetry in Fig. 2. In the absence of proper refreezing measurements, the good performance of the 1 km SMB data set gives confidence in the downscaled refreezing product since it is computed as a residual between realistically modelled SMB components (runoff, melt and precipitation).

Notwithstanding your rather successful comparisons to observed mass balance, the fact remains that Eq8 is the end of a long chain of educated guesses. As a minimum it needs more discussion. Why is all rainfall assumed to refreeze?

Rainfall is only refrozen when it falls on freezing snow; we included rainfall in Eq. 8 merely to close the liquid water balance.

Nearer to the start of the chain, how does RACMO2.3 calculate melt ME and runoff RU?

In RACMO2.3, melt is calculated by closing the surface energy budget at the snow/ice surface as:

$$ME = SW_{net} + LW_{net} + SHF + LHF + G_s$$

where ME is total melt energy available at the surface, SW_{net} is the net shortwave radiation, LW_{net} is the net longwave radiation, SHF and LHF are the turbulent sensible and latent heat fluxes and G_s is the sub-surface heat transfer.

The resulting meltwater and rain can percolate through snow layers and fill the available pore space to contribute to refreezing until the whole firn/snow column becomes saturated. Additional melt or rain will run-off to the ocean. To clarify, we inserted the following lines in the Supplementary Information: “In RACMO2.3, the excess energy available at the surface, resulting from closure of the surface energy budget, is used to melt snow and ice. Meltwater and rain percolate through the firn column and progressively fill the pore space until the entire firn column is saturated. At this point, any additional water is assumed to run off.”

Stylistic Comments:

P1

L1: Delete “contiguous”. Same at L17, L27-28 and so on. It is unnecessary anyway, but is also ambiguous because nearly a quarter of the peripheral-glacier area (ref. 23’s CL1 glaciers) is “contiguous” with the ice sheet. OK.

L9: Surely “pre-1997” should be “post-1997”? **OK.**

P2

L15: Change “glaciated regions” to “glacierized area”. **OK.**

L26: Why is “changes in” in parentheses? I suspect that it was marked for deletion but not deleted. **OK, removed.**

P3

L39: “entire Greenland” is not idiomatic English. You have to say “all of” or “the whole of”. **Thank you, changed.**

L42: “w.e.” is the preferred abbreviation for “water equivalent”, and you should not omit the essential “a-1” – either here or in later instances of the unit. **Thank you, changed.**

L46: Change “later in the main text” to “below”. **OK.**

L47: Insert “observations” before “over”. **We inserted “measurements” instead of observations.**

L48: What does “(v1.0)” refer to? **This is the current version of the downscaled product. Removed.**

P4

L64: Unfortunately you cannot translate German Haushalt as “household”, which is English for a house and its contents (including its human occupants). You have to say “balance” or “budget”. (I am afraid I do not know the Dutch for the German word.) **We replaced household by “balance”.**

L68: “for maintaining”. **OK, changed.**

P5

L81-82: “Tables S1 and S2”; but follow journal style about abbreviating “Table” as at L82. “responses”. **Thank you, changed.**

L86-87: Soften “eventually leading to” to “implying eventual”. Change “refreezing” to something like “meltwater retention”. **Thank you.**

P6

L91-93: Change “Fig. S4 contrasts” to “we examined”, and cite Figure S4 at the end of the sentence. **OK.**

L97: “twice as large ... as for southern”. **OK.**

L105: “zones”. **OK.**

L110: Change “deteriorated” to “reduced”, and preferably “refreezing” to “meltwater retention [or ‘storage’]”. **OK, we kept refreezing capacity in the revised manuscript.**

P7

L120: “correlation with”. **OK.**

L124-125: “for the dependence of modelled SMB components on the 11-km”. **OK.**

L126: Change “glaciated” to “glacierized”. **OK.**

L126-127: The current pixel is not adjacent to itself, and has already made its appearance a few words earlier. So change “6” to “five” and delete “including the

current one". We reformulated accordingly.

L130: Spell out "three". OK.

L139: The right-hand side of Eq3 would be more readable if it ran "... 0.5 (1+ ξ) SWdirect 1 km/Lf ". See response to reviewer #1.

L141: "(-)" slowed me down by several seconds; expand it to "(dimensionless)", or just leave it out. We replaced by "(dimensionless)".

L144: Italicize Lf. " ξ is the dimensionless ...". Could you add a short explanation of why ξ is needed, and where you get the tilt (slope?) from?

The glaciers slope and orientation were derived from the topography of the GIMP DEM at 1 km. We reformulated as: "To account for the slope of the glacier surface, the dimensionless correction factor for a tilted plane ξ is applied to the direct component of modeled downward shortwave radiation. This correction is required as RACMO2.3 models radiation assuming a horizontal surface. This factor ranges from 0 for north sloping glaciers to values larger than 1 for south oriented slopes."

L146: Delete the unnecessary "specific", and perhaps change "fraction" to "ratio". You could then change the period to a colon and delete L147. Italicize Γ at L146,148.

We reformulated as: "A daily ratio Γ , i.e. between downscaled runoff and melt in v0.2, is applied to MEadd to calculate the additional runoff (RUadd)."

P9

L150: "ascribable". OK.

L151: Italicize f here and in Eqs 5,6. "least-squares". OK.

L152: Change "reason" to "justification". OK.

L166: "daily precipitation totals from v0.2 were". OK.

P10

L177: You cannot perform a method. Say "we used". OK.

P12

L215: z..?? We updated the reference.

L219: Do not capitalize the paper title. OK.

P15

L262: "by preparing mask files and improving figures". OK.

Fig1

L4: "five".

L5: "gradients are" should be "is" (there are no fitted lines). "downscaled simulations (bmod; blue dots) and in-situ data(bobs; red dots)". Panel b: explain b0 and b1. Oddly, b0 seems to be the slope and b1 the intercept of the regression line. OK. We inserted the following equation in the caption: "The red dashed line represents the regression including all measurements ($y = b1 + b0 * x$).

Fig3 L4: I think "integrated annual mean" should be "annual". "integrated" is redundant, and "mean" suggests that you divided by the number of days or something, when actually you just summed over the year. Thank you.

Fig4 Delete "mean" at L3, but not at L6. OK.

Supplementary Information:

P2

L8: “glacierized”. OK.

L17: What is the “original” product? The GIMP DEM? If so, you have to distinguish between the averaging of elevations and whatever was done (possibly nothing) to assign “glacier” codes in the 1-km mask.

We clarified this by reformulating as follow: “by averaging the original GIMP DEM at 90-m resolution.” The PROMICE ice classes were projected on the down-sampled version of the 1 km ice mask derived from the GIMP DEM (nearest pixel approach).

L18: Delete “contiguous”. OK.

L19: As in the main text, you should say explicitly here that the CL2 glaciers are treated as part of the ice sheet.

We clarified this by inserting “(including connectivity level CL2 in Ref.23)” at **L19**.

L27: “albedos”. OK.

P3

L39: “grid cell”. OK.

L40: Delete the redundant “adjacent”. OK.

TableS1 and S2: Use the same symbols for balance components as in Eq1 of the main text. For consistency, we replaced P_{tot} in Eq1 and at **L120** by “PR”.

TableS3: The references should probably carry the same numbers as in the main text, but once again I am baffled by the mention of Cogley 2009 (see comment above at P2 L20). Here, however, we have confirmation that you are actually citing Marzeion et al. 2012, ref. 15; Cogley 2009 (as corrected above) was the source for in-situ observations in ref.15, but only as a paper documenting a dataset that was updated for ref. 15. We removed reference 13 accordingly and replaced it by reference 15.

L4: “Uncertainties ... are obtained”. But see comment on the main text (P4 L54). See our response to question at **L52-54**.

FigS1: In panels b and c the dazzling red of the ice sheet tends to obscure the information you are trying to convey; perhaps light grey instead as in FigS4?

We used light grey over the ice sheet in the revised manuscript.

FigS2: “by regression against elevation”. We updated Supplementary Figure 2 based on Noël et al. (2016). We also included the following caption: “Elevation-dependent downscaling procedure: b_{11km} and a_{11km} are respectively the daily local estimates of the SMB components regression to elevation and the SMB components value at mean sea level obtained on the RACMO2.3 grid at 11 km. The red line corresponds to the regression (b_{11km}) calculated using the current grid cell (blue dot) and the adjacent ones (red dots). The dashed green line applies the regression slope to the current grid cell to estimate a_{11km} (from Noël et al., 2016).”

FigS4 L2: delete “mean”. OK.

L3: “The changes ... are”. Thank you.

REVIEWERS' COMMENTS:

Reviewer #1 (Remarks to the Author):

I looked into the revised manuscript and response letter. All my previous comments have been successfully addressed and I have no further objections. I particularly like the addition of Figure S4 and the related description of elevation profiles of refreezing.

Reviewer #2 (Remarks to the Author):

Comments on "A tipping point in refreezing accelerates mass loss of Greenland's glaciers and ice caps", by B. Noël et al., submitted to Nature Communications
Graham Cogley, December 2016

General Comments

This revision of a text that I reviewed earlier is very thorough and on the whole satisfactory. The error model, for example, is a significant improvement (L52-54 in both the original and revised texts). The authors have not accepted several of my points, particularly some of those about clear terminology, but I have to acknowledge that the terms and practices they have retained against my advice are fairly widespread in the ice-sheet modelling community.

This does not mean that I accept any of their rebuttals. To illustrate, they have missed the unstated point of my objection to cumulative mass balance in Figure 2 instead of the mass-balance rate (L49-50). There is scope for debate about what ICESat and Cryosat-2 measure – you could refer each new altimeter reading to its immediate collocated predecessor just as easily as to an initial state. But my point is that cumulative plots are a visual suppression of the noise that the authors say they do not want to introduce, and that this offers the reader a possibly misleading sense of confidence. But we may have to agree to disagree on this.

I noted only a few small points requiring clarification in the revision:

L44 There should be a space between "mm" and "w.e.", here and throughout. At one level this is a trivial matter: "Who cares?" But in a broader view it is worthwhile to follow standards, and here the applicable standard is the Glossary of Glacier Mass Balance and Related Terms (<http://unesdoc.unesco.org/images/0019/001925/192525e.pdf>; page 16). "mmw.e." is just as wrong as "Km" for "km". (It is true that I chaired the IACS Working Group that produced the Glossary, but it was a large collaborative effort across the field of glaciology.)

L55 Change "glaciated" to "ice".

L125 "interior" should perhaps be replaced by "upper".

L170 "i.e." is unnecessary.

L201 "zone".

In the Supplementary Information:

L12 "progressively fill the pore space": can you clarify this a little? E.g. surely you do not mean "progressively from the top down". Do you simply use the meltwater to increase the density of the bulk firn layer until it reaches 900 kg m⁻³?

L61 "the GICs".

L67 "Landsat satellite imagery (Google Earth)": which? They look like Landsat to me.

P9ff. It would be helpful to remind the reader of the definitions of the acronyms in the caption of Table S1. The specific-unit rows in these tables are welcome.

Fig. S1 Edit out the curious grey triangle in the top right of panel a.

Author's response.

Reviewer #1 (Remarks to the Author):

I looked into the revised manuscript and response letter. All my previous comments have been successfully addressed and I have no further objections. I particularly like the addition of Figure S4 and the related description of elevation profiles of refreezing.

Reviewer #2 (Remarks to the Author):

General Comments

This revision of a text that I reviewed earlier is very thorough and on the whole satisfactory. The error model, for example, is a significant improvement (L52-54 in both the original and revised texts). The authors have not accepted several of my points, particularly some of those about clear terminology, but I have to acknowledge that the terms and practices they have retained against my advice are fairly widespread in the ice-sheet modelling community.

This does not mean that I accept any of their rebuttals. To illustrate, they have missed the unstated point of my objection to cumulative mass balance in Figure 2 instead of the mass-balance rate (L49-50). There is scope for debate about what ICESat and Cryosat-2 measure – you could refer each new altimeter reading to its immediate collocated predecessor just as easily as to an initial state. But my point is that cumulative plots are a visual suppression of the noise that the authors say they do not want to introduce, and that this offers the reader a possibly misleading sense of confidence. But we may have to agree to disagree on this.

To stress the fact that we use cumulative mass change to suppress the random noise from ICESat/Cryosat-2 data, we included the following sentence in the Methods section of the main manuscript: “Fig. 2 shows the cumulative GICs mass change; using cumulative values suppresses noise in the ICESat and CryoSat-2 time series”.

I noted only a few small points requiring clarification in the revision:

L44: There should be a space between “mm” and “w.e.”, here and throughout. At one level this is a trivial matter: “Who cares?” But in a broader view it is worthwhile to follow standards, and here the applicable standard is the Glossary of Glacier Mass Balance and Related Terms (<http://unesdoc.unesco.org/images/0019/001925/192525e.pdf>; page 16). “mmw.e.” is just as wrong as “Km” for “km”. (It is true that I chaired the IACS Working Group that produced the Glossary, but it was a large collaborative effort across the field of glaciology.)

We replaced “mmw.e.” by “mm w.e.” throughout the main text, Supplementary Information, Figures legend and captions.

L55: Change “glaciated” to “ice”. OK.

L125: “interior” should perhaps be replaced by “upper”. OK.

L170: “i.e.” is unnecessary. OK.

L201: “zone”. OK.

In the Supplementary Information:

L12: “progressively fill the pore space”: can you clarify this a little? E.g. surely you do not mean “progressively from the top down”. Do you simply use the meltwater to increase the density of the bulk firn layer until it reaches 900 kg m^{-3} ? We clarified this by reformulating as: “Liquid water from melt and rain percolates through the firn column, and is either held as irreducible water or refreezes, progressively reducing pore space from bottom to top layers until the entire firn column turns into ice (900 kg m^{-3}) and no additional water can be stored.”

L61: “the GICs”. OK.

L67: “Landsat satellite imagery (Google Earth)”: which? They look like Landsat to me. The background of Supplementary Figure 1 stems from Landsat satellite imagery that we retrieved from Google Earth. We decided to remove “(Google Earth)” and to insert the following statement in data availability: “The background in Supplementary Figure 1 stems from Landsat satellite imagery freely available at <https://landsat.usgs.gov/>.”

P9ff: It would be helpful to remind the reader of the definitions of the acronyms in the caption of Table. We included the following sentence in the caption of Tables S1 and S2: “SMB components include runoff (RU), total precipitation (PR), total melt (ME) and refreezing (RF).”.

S1: The specific-unit rows in these tables are welcome. OK.

Fig. S1: Edit out the curious grey triangle in the top right of panel a. OK. Thank you.